# Deep phenotypic profiling of neuroactive drugs in larval zebrafish

Leo Gendelev [1], Jack Taylor [1,2], Douglas Myers-Turnbull[1], Steven Chen[3], Matthew N. McCarroll[1,3], Michelle R. Arkin [3], David Kokel[1] ✉ & Michael J. Keiser [1,3,4,5] ✉

Behavioral larval zebrafish screens leverage a high-throughput small molecule discovery format to find neuroactive molecules relevant to mammalian physiology. We screen a library of 650 central nervous system active compounds in high replicate to train deep metric learning models on zebrafish behavioral profiles. The machine learning initially exploited subtle artifacts in the phenotypic screen, necessitating a complete experimental re-run with rigorous physical well-wise randomization. These large matched phenotypic screening datasets (initial and well-randomized) provide a unique opportunity to quantify and understand shortcut learning in a full-scale, real-world drug discovery dataset. The final deep metric learning model substantially outperforms correlation distance–the canonical way of computing distances between profiles–and generalizes to an orthogonal dataset of diverse drug-like compounds. We validate predictions by prospective in vitro radio-ligand binding assays against human protein targets, achieving a hit rate of 58% despite crossing species and chemical scaffold boundaries. These neuroactive compounds exhibit diverse chemical scaffolds, demonstrating that zebrafish phenotypic screens combined with metric learning achieve robust scaffold hopping capabilities.

The mechanism of action of central nervous system (CNS) drugs remains poorly understood, even for those used for decades (e.g., ketamine[1]). The complex nature of G-Protein Coupled Receptor (GPCR) and ion channel-mediated pathways of the vertebrate nervous system[2–11] exacerbate the problem. Because of the prevalence of polypharmacology in neuroactive drugs[12], a magic bullet single-target approach to drug discovery[13] falls short[14]. Phenotypic screening circumvents these problems by identifying compounds that may interact with individual or multiple targets[15,16]. These screens prioritize desired and often biologically complex readouts of induced phenotypes on higher-level model systems. Despite historically limited throughput, rapid phenotypic profiling of thousands of compounds in vivo is now possible using larval zebrafish[17–21]. These vertebrates have high levels of shared genetics[22,23] and CNS anatomy[24] (with humans) and scale to high-throughput testing of complex behavioral readouts[2–11]. Phenotypic screening in larval zebrafish, combined with human-target-based cheminformatic methods such as the Similarity Ensemble Approach (SEA)[25,26], and enrichment factor (EF) calculations[10,27–29], have enabled drug discovery and target deconvolution for neuroactive phenotypes in mammals.

However, high-content zebrafish behavioral screening data are both a blessing and a curse for pharmacological studies because of the challenges in extracting and comparing features in the collected video data. In previous work, larval zebrafish, plated on 96-well plates, were

[1]Institute for Neurodegenerative Diseases, University of California, San Francisco, San Francisco, CA, USA. [2]UCSF Weill Institute for Neurosciences Memory and Aging Center, University of California, San Francisco, CA, USA. [3]Department of Pharmaceutical Chemistry, University of California, San Francisco, San Francisco, CA, USA. [4]Department of Bioengineering and Therapeutic Sciences, University of California, San Francisco, CA, USA. [5]Bakar Computational Health Sciences Institute, University of California, San Francisco, San Francisco, CA, USA. ✉e-mail: dave.kokel@gmail.com; keiser@keiserlab.org

treated with various compounds, and various stimuli − including acoustic stimuli and high-intensity light of different colors − elicited a broad spectrum of behavioral responses in the fish[10,28]. Videos recorded each well, from which a "motion-index" (MI) time series is computed to measure bulk motion over time (Fig. 1a–d[7], Fig. 1b and Equation 1). Traditionally, the phenotypic distance between MI time series is computed using correlation distance[28]. Other approaches have included classification and video analysis using machine learning[2,30–34]. Correlation distance reliably discriminates antipsychotic[28] and anesthetic[10,28,29] phenotypes but fails to distinguish

more subtle phenotypes. Indeed, fish rarely respond to stimuli in a one-to-one video frame correspondence when each frame is 1/100th of a second, breaking a basic assumption of how these MI distances are traditionally computed. In an experiment with various assays, the strength of the response to each assay may vary in drug-treated fish; however, correlation distance values all frame contributions equally.

Various deep learning and unsupervised learning techniques have been employed toward zebrafish behavior classification[35]. We sought a distance metric that leveraged zebrafish phenotypic screens for a broader range of induced behaviors. Specifically, we used a class of

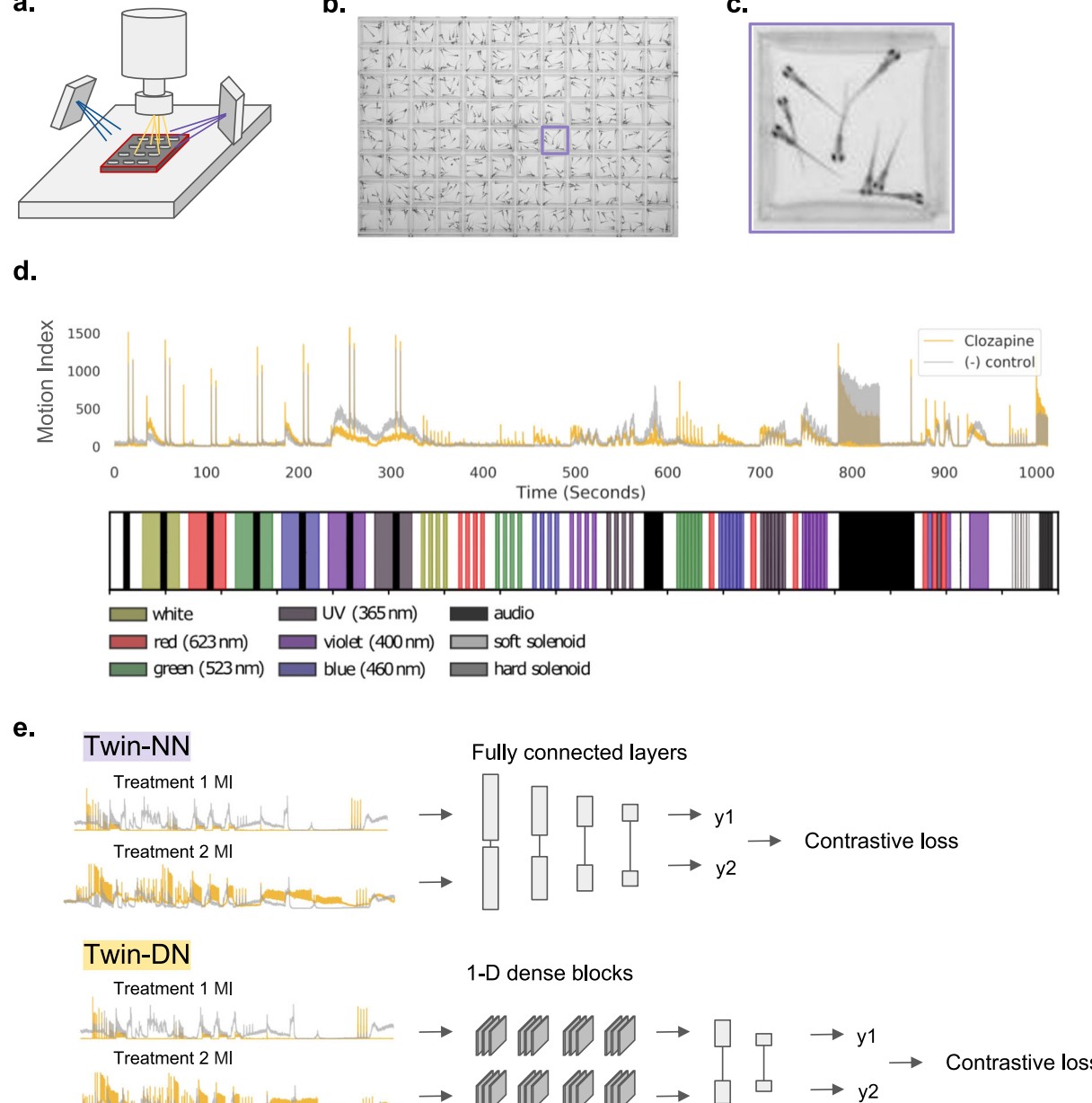

**Fig. 1 | Zebrafish behavioral screening and architecture.** Diagrammatic representation of zebrafish experimental screening setup, motion index calculations, and deep metric learning model architectures. **a** Simplified representation of a zebrafish screening platform, with larval zebrafish in 96-well plates under a camera subject to varied stimuli such as blue light, purple light, acoustic stimuli, and physical tapping. **b** Example of a representative video frame. **c** Zoom view of a single well. **d** Example motion index (MI) time series for clozapine-treated (gold) fish and negative control (DMSO) (gray) wells. The MIs are averaged across all the drug- and control-treated well replicates. A stimulus legend shows the stimuli types occurring at different times in the assay. **e** Deep metric learning model architectures: Twin-NN (top) and Twin-DN (bottom). In both models, the input is a *pair* of MI time-series vectors passing through multiple neural net layers. A contrastive loss function[72] scores the two learned output vectors (y1 and y2) distances based on whether the input MI vectors were from the same or different treatments. As in (**d**) above, gold denotes drug-treated MI, and gray is the negative control (DMSO). Source data are provided in the Source Data File.

neural networks uniquely suited for learning distances between pairs of inputs (Siamese neural networks, or twin neural networks, twin-NNs, Fig. 1e). These models were initially developed for biometric finger-print verification[36], subsequently finding use in many machine learning (ML) tasks, such as "one-shot learning" on small datasets for image classification[37,38]. Siamese networks pre-date and often became the neural network models and loss functions subsequently adopted by various studies in deep metric learning and contrastive learning. Siamese networks have been applied to study zebrafish behavior at lower throughput[39] or at higher throughput for embryonic development[40], but to our knowledge, not in the setting of high-throughput larval zebrafish phenotypic screening.

In this work, we screen a library of 650 ligands (from the SCREEN-WELL Neurotransmitter Set, "NT-650", Methods) in high-replicate and train twin-NNs to relate drugs via the phenotypes they induce in larval zebrafish. We construct the screens from the ground up with ML model training in mind, but the models still exploit unanticipated artifacts in the resulting screen dataset via an undesirable process known more broadly as "shortcut learning"[41]. We initially study the effect of retraining the deep metric learning models on synthetically randomized datasets that we design to test for confounding effects, ultimately driving the redesign and re-collection of a new experimental screen with full physical randomization. Models trained on the revised screen cluster diverse neuroactive compounds in a way that corresponds strikingly well with known neuroactive biology, and they phenotypically link structurally distinct compounds by scaffold hopping[42,43]. Finally, the learned distance metric generalizes to a screening dataset of diverse drug-like compounds unseen during training, automating the discovery of neuroactive compounds active on human receptors when tested prospectively in vitro.

## Results

### Twin neural networks identify drug replicates from complex behavioral readouts

We collect a high-throughput phenotypic dataset based on the NT-650 neurotransmitter library screened in high replications (7–10 replicates per drug) for training machine learning models. We plate larval zebrafish onto 96-well plates (8 fish per well) and dose wells with drugs at a 10 μM concentration, a reasonable dose for in-vivo primary screening of neuroactives (Fig. 1a–c). Various stimuli, such as acoustic sounds, light stimulus, and physical tapping of the multi-well plate stage, are performed to elicit diverse behavioral responses in the fish, as optimized previously[7] (Fig. 1d). We record videos of the fish's behavior throughout the experiment. For each well, we encode and convert videos of larval fish into aggregate motion over time, resulting in a time-series vector, or motion index (MI).

We evaluate how well twin NNs can identify whether two MI profiles, such as those shown in Fig. 1e, originate from the same category – specifically, whether they are caused by the same drug. This is in contrast to other correlation metrics that often fail to reliably recognize when different samples have been affected by the same drug, especially when the resulting phenotypic changes are subtle. By necessity, a Twin-NN must learn which time points are most informative and how to correct for slightly or partially misaligned MI traces to correctly group same-drug replicates accurately across a diverse range of pharmacology and their concomitant behavioral traces. Twin NNs consist of twin encoding layers, which share model weights and operate on a pair of different inputs (MI traces) to output a distance reflecting whether the MIs represent replicates of the same drug (distance = 0) or traces from mismatched drugs (distance > 1)[36].

We filter the dataset to remove human drugs that do not alter zebrafish behavior, namely those whose MI traces cannot be distinguished from vehicle controls with a simple random forest model (see "Methods", Supplementary Fig. 1). Drugs can fail to induce strong behavioral responses in zebrafish due to many factors, including

differences in cross-species biology, concentration, incubation time, absorption route, or other factors. The neural network embedding architecture of each half of the model is a design choice; we implement a fully connected multi-layer perceptron (Twin-NN) (Fig. 1e) as a baseline model and a second architecture motivated by DenseNet[44] (Twin-DN) as a more computationally expressive alternative. We considered using recurrent architectures – neural networks designed to operate on sequences, such as the LSTM[45] or GRU[45,46] – but were concerned that the long length of the time series samples raised issues with vanishing gradients and run-time.

Prior work using zebrafish behavioral MI for scaffold-hopping and phenotypic drug discovery predominantly uses vector distance calculations to compare MI traces without warping, alignment, or relative weighting of individual time points. Compared to these conventional correlation and Euclidean distance approaches, both the Twin-NN and Twin-DN models discern positive (matched) and negative (mismatched) drug replicate pairs with drastically improved performance (Fig. 2a), with Twin-DN scores achieving 0.97 ROC-AUC and 0.98 PRC-AUC (Fig. 2b, c). We observe a near-perfect ability of the learned distance metric to discern MIs of replicates from the same drug from MIs of different drugs. Further, a Uniform Manifold Approximation and Projection (UMAP)[47] plot (Fig. 2d), calculated over the means of the time series for all replicates of all 650 drugs in the screen, yields pronounced, discrete, and localized clusters. While heartening, this performance was substantially higher than we had anticipated and to a suspicious degree: many compounds do not reliably induce larval zebrafish behavioral phenotypes. Nevertheless, these results suggested that nearly all the compound library's experimental replicates could be grouped by the Twin-DN model with near-perfect fidelity. We wondered whether the model's exceptional performance might rely instead on shortcut learning[41] or the exploitation of hidden artifactual cues encoded within the data that were invisible to human researchers but perceivable by the deep learning model.

### Machine learning exploits high-frequency components and plate-location effects

We evaluate the presence of shortcut learning in our model by testing how well the learned distance metric generalizes to an archival quality control (QC) screen ("Methods") performed at an earlier time on data never seen by the metric-learning models. For this screen, we select 14 drugs with diverse mechanisms of action (MOAs) and assay them, also in high replications, along with a vehicle (dimethyl sulfoxide; DMSO) control and lethal control (eugenol). To test generalizability, we train a k-nearest neighbors classifier on the QC replicates using one of three distance metrics: Twin-NN, Twin-DN, or correlation distance ("Methods"). For most drugs, the Twin-DN distance metric underperforms correlation distance (Fig. 2e, orange bars). We suspected that the expressive Twin-DN models readily memorize the high-frequency components in the time series, which may come from artifacts such as plate vibrations or high-frequency noise in the imaging sensor. Indeed, when we ablate the high-frequency components of the time series (with a Hanning smoothing filter[48] as implemented in the scipy package[49]), Twin-DN performance drops precipitously on the original NT-650 screen, with Twin-NN performance likewise dropping, but to a lesser extent (Fig. 2c, d). These results indicate that the learned distance metrics exploit the high-frequency components of the NT-650 screen data and that these hidden "shortcut" patterns do not generalize to the separate QC screen.

Per standard practice, we had already attempted to address potential overfitting or data leakage by splitting the training and validation sets by drug - leaving no replicates of any one drug in common between train and validation splits. In addition, we performed a set of further machine-learning soundness checks (scrambling data labels and randomizing the training data features (Supplementary Fig. 2), so the Twin-DN model's evident exploitation of high-frequency signal was

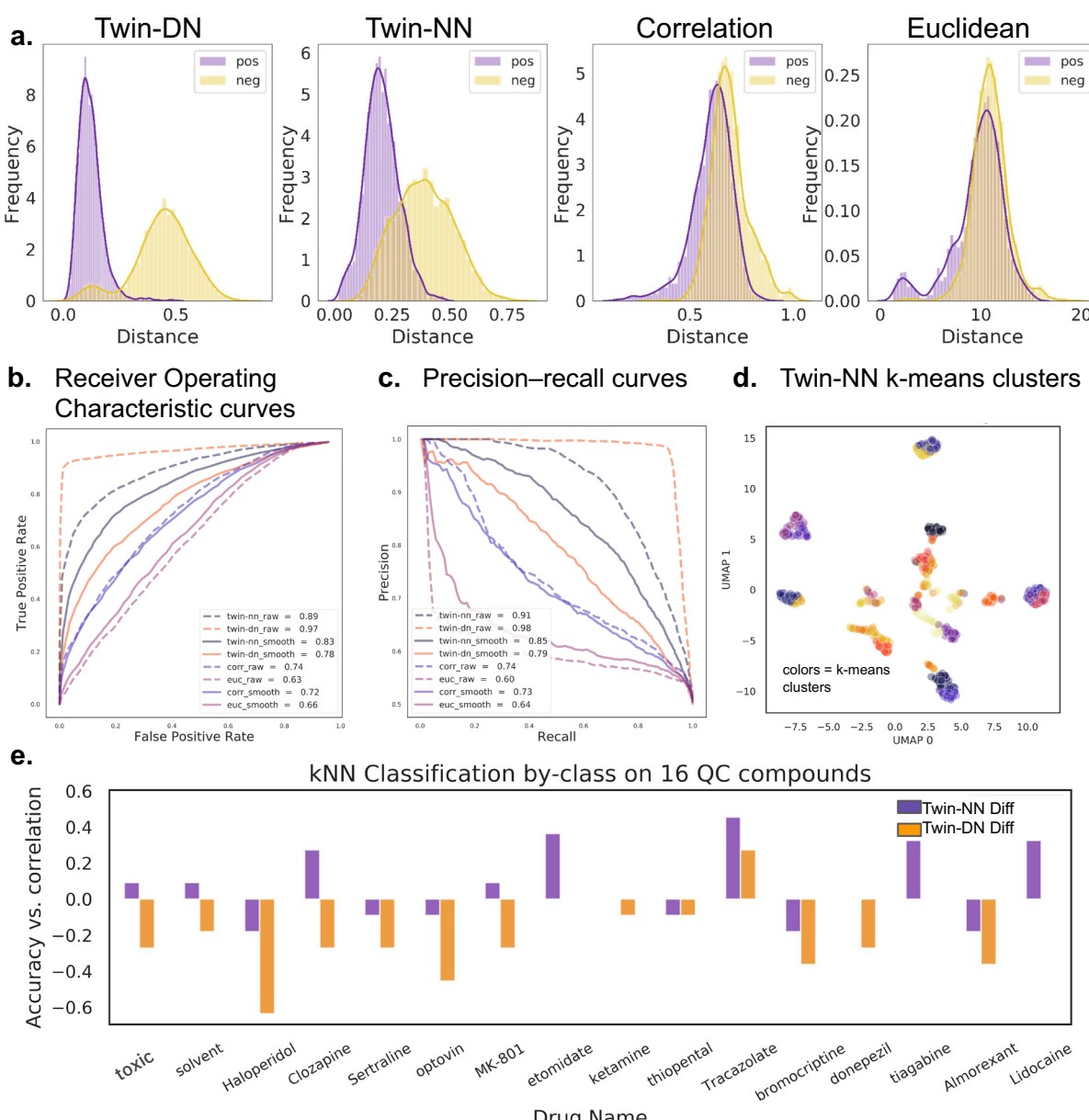

**Fig. 2 | Metric learning models exploit high-frequency components of time-series signals in an initial non-randomized screen.** We compare the Twin-NN and Twin-DN models against traditional methods such as correlation and Euclidean distance on both raw and smoothed motion index (MI) time series data and examine how well the models cluster drugs and generalize to a separate dataset not used for training. **a** Separation of positive (treatment replicate) and negative (mismatched replicates) MI vector pairs using the Twin-DN model (left), the Twin-NN model (2nd column), correlation distance ("Correlation," 3rd column), and Euclidean distance ("Euclidean," right). The Twin-DN and Twin-NN models exhibit a drastically improved ability to separate positive from negative pairs, as evidenced by the strong distance separation (x-axis) between the positive and negative pair distributions. Euclidean and correlation distances fail to separate the same- from mismatched replicates for most MI pairs, except for those with minimal distances (e.g., the most phenotypically similar pairs). **b** Receiver operator characteristic plots. Twin-DN achieves the best area under the curve (AUC = 0.97), followed by

Twin-NN (0.89). Performance dropped drastically for both learning models using MI time-series inputs smoothed with a Hanning window[48] of size 11, particularly Twin-DN (from 0.97 to 0.78). Correlation and Euclidean distance were robust to Hanning smoothing. **c** Precision recall curves, showing trends consistent with (**c**). **d** A UMAP[47] using the Twin-DN distances reveals extreme clustering with distinct phenotypic islands; as many drugs are unlikely to induce strong phenotypes in the fish, this was an unexpected and suspicious result. **e** We trained k-Nearest-Neighbor (kNN) classifiers using scikit learn[63] on a separate high-replicate MI trace dataset of 16 quality control drugs never used for training or model evaluation. For many drugs (e.g., haloperidol), the Twin-DN-based distance underperforms the zero-baseline defined by kNNs using correlation distance. Twin-NN distance outperforms correlation distance on a few drugs (e.g., tiagabine and lidocaine) and always matches or exceeds the Twin-DN model. Source data are provided in the Source Data File.

initially surprising. However, all compounds in the NT-650 library arrived from the supplier in preset layouts, meaning all replicates of the same compounds (or all positive pairs) in the dataset always corresponded to identical plate locations. In contrast, mismatched pairs could come from any combination of compound locations across and within the plates. Our original experimental design for the screen did

not control for the potentially confounding layout effect. Even a simple machine-learning architecture might be able to learn light-based patterns for distinguishing different location pairs. The Twin-DN model performance took the biggest hit with data smoothing, suggesting that the more expressive a model is, the more readily it can exploit feature shortcuts.

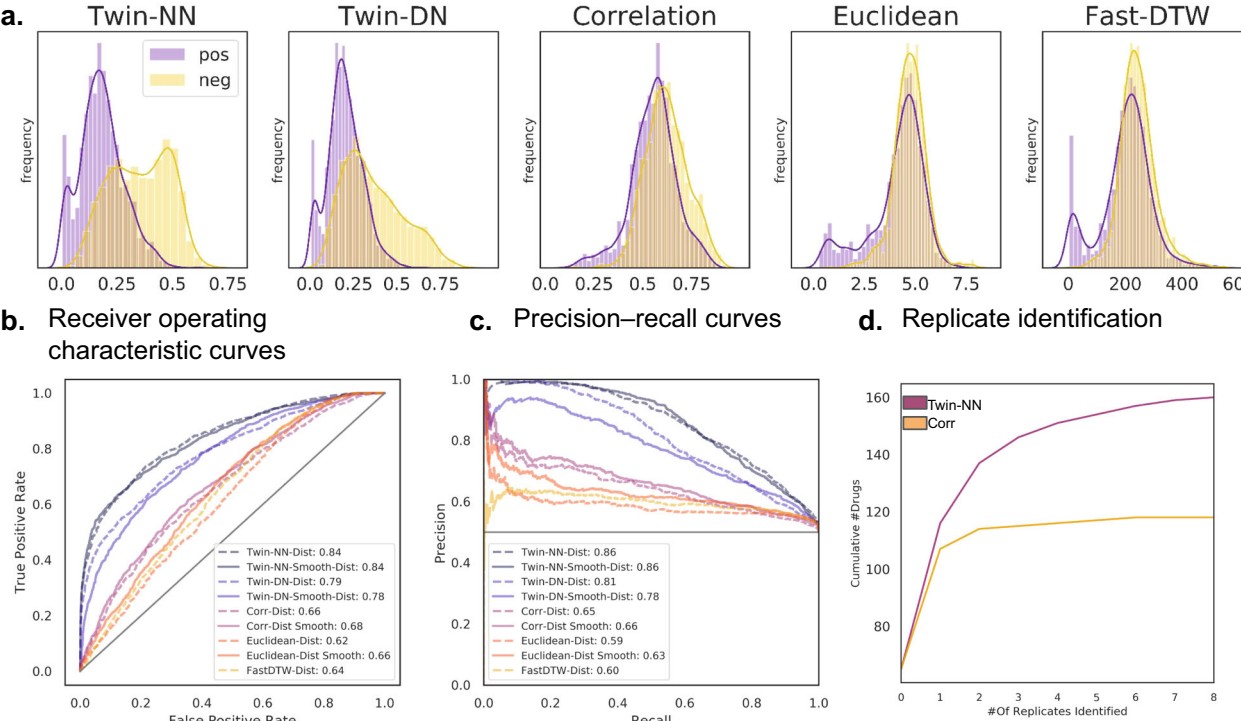

**Fig. 3 | Metric learning operates actionably on a fully randomized screen.** We investigate models trained on the second, fully randomized screen. **a** Separation of positive and negative motion index (MI) trace pairs from the fully randomized screen with Twin-NN (left), Twin-DN (2nd column), correlation (3rd column), euclidean (4th column), and Fast-DTW (right) distances. Assessed as in Fig. 2a, the revised deep learning models significantly outperform correlation, euclidean, and fast-DTW distances. **b** Twin-NN and Twin-DN receiver operator characteristic performance is similar (AUC = 0.84 and 0.79, respectively) and significantly exceeds correlation, euclidean, and fast-DTW (0.66, 0.62, and 0.64). Notably, models trained with and without Hanning smoothing no longer differ significantly. **c** Precision recall curves are consistent with (**b**). **d** The Twin-NN model identifies

matched drug replicates more effectively than correlation distance, which typically starts to fail beyond one replicate. We compute an all-by-all distance matrix across NT-650 compounds at the individual replicate level and determine how many replicate wells of the compound appear within the top 50 most similar ranked wells. We plot the cumulative total of unique drugs (y-axis) versus the increasing count of identified replicates (x-axis). The y-axis maximum does not reach the total number of NT-650 compounds because neither method perfectly ranks all replicates within the top 50 most phenotypically similar ranked wells for all NT-650 compounds. Indeed, some compounds are inactive, with replicates indistinguishable from DMSO. Source data are provided in the Source Data File.

## An optimized experimental screening design

To unequivocally control for within-plate positional confounding effects, we perform a second high replicate screen of NT-650, but this time with the treatments fully robotically randomized across plates and wells ("Methods"). We also include wells treated with a high dose of the anesthetic eugenol as a control[7] baseline for lethality. We take a near-identical approach to pre-filter for drugs without effect as in the original screen, except that we use the random forest model to label the MI profile into three possible bins: "active," "inactive," and "lethal" ("Methods", Supplemental Fig. 1).

We train new Twin-NN and Twin-DN models on this experimentally randomized NT-650 dataset (NT-650-revised). While the SNN models achieved slightly lesser performance on the randomized dataset than on the original non-randomized (NT-650-naive) well layout dataset (e.g., 0.84 vs. 0.89 AUROC for Twin-NN and 0.84 vs. 0.97 AUROC for Twin-DN), their performance still dramatically exceeded that of correlation distance and Euclidean distance approaches (0.66 and 0.62 AUROC, respectively). Striking differences in the distribution of Twin-NN and Twin-DN distances for the positive and negative pairs (Fig. 3a) agree with greater ROC-AUC and PRC-AUC performance (Fig. 3b, c). Fast-DTW, a popular dynamic time-warping approach for time series prediction that optimizes the alignment between time series[50], marginally improves on Euclidean distance and falls short of correlation distance in classifying positive versus negative pairs. Training these baseline models on the NT-650-revised screen with computationally smoothed high-frequency components did not

significantly reduce performance, indicating the models no longer rely on high-frequency feature components.

Another way of assessing model performance is by measuring its ability to identify replicates of a compound. In the ideal case, a model can identify all replicates for compounds that induce significant behavioral responses. In practice, experimental replicates will often be ineffective for many reasons. However, the better a model performs, the more replicates across drugs it can identify. Although correlation distance identified one replicate for most drugs, it rarely identified three or more replicates, whereas Twin-NN did so frequently and sometimes picked up all 7–8 replicates (Fig. 3d). This effect is emphasized by the early plateauing of the cumulative count curve for correlation distance. The total cumulative count does not reach the total number of drugs for either method because no method perfectly identified all replicates for all drugs, some of which may have been inactive and thus indistinguishable from negative control (DMSO).

## Mapping a larval zebrafish "behaviorome"

Using the Twin-NN learned distance metric, we cluster the compounds' MI traces from the fully randomized NT-650 screen and visualize the resulting phenotypic landscape by UMAP[47] (Fig. 4a). We observed defined clustering and structure within this view, representing a behavior-based pharmacological map of 650 known human drugs in larval zebrafish. Ineffective drugs populated the yellow region, while drugs inducing behavioral readout changes favored the violet region. The most robust phenotypes appeared towards the bottom of the plot,

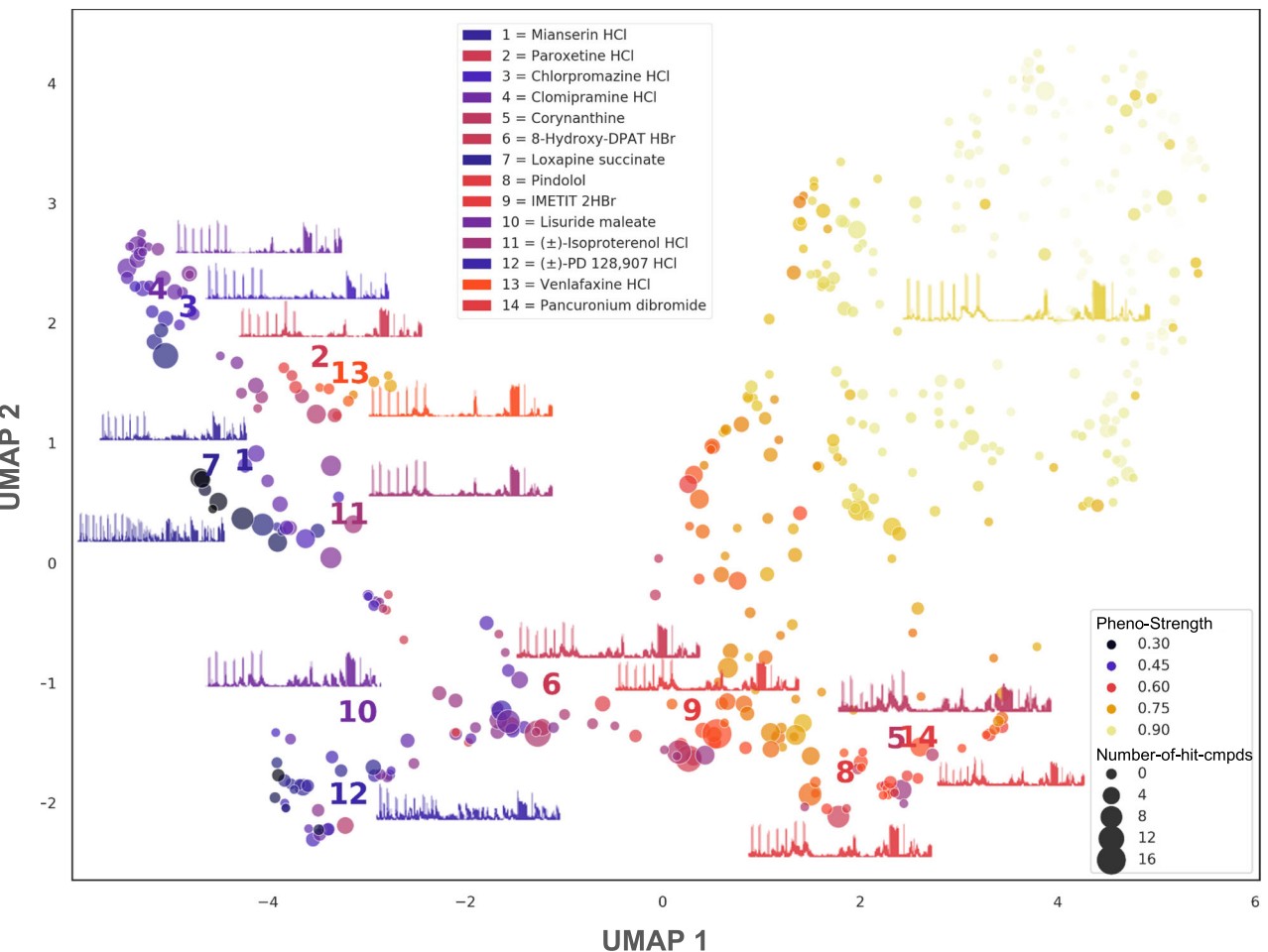

**Fig. 4 | A learned phenotypic distance identifies islands of drugs by protein target profile that correlation distance cannot.** We investigate how well the learned phenotypic distance meaningfully clusters drugs using a UMAP[47] on Twin-NN distances between the average time series across all replicates of each drug of the fully randomized NT-650 screen. Labeled example drugs represent anchor points across the phenotypic landscape. Dot color changes by phenotype strength as determined by a separate random forest classifier employed earlier in the dataset construction process ("Methods"). A UMAP on correlation distances of the same data (Supplementary Fig. S3) fails to form meaningful phenotypic clusters. Source data are provided in the Source Data File.

falling in the negative value range for principal component 2 (*y*-axis) of the UMAP. When we colored the 325-compound subset predicted to have strong phenotypes ("Methods") by the generic functional classes corresponding to plate-wide assignments by the chemical vendor (e.g., "opioids", "serotonergic ligands"), we observed only a weak correlation with the position in the UMAP (Supplementary Fig. 16). This was unsurprising, as these classes do not account for the poly-pharmacology of many neuroactive compounds.

One qualitative way to assess the behaviorome layout is whether specific drugs with similar known MOAs and indications group together. We highlight several such drugs in this plot with labels. The SSRIs fluoxetine and paroxetine clustered (labels 2,13) but distinctly separated from the tricyclic antidepressant clomipramine (label 4), although these shared a broader neighborhood as expected. This observation is consistent with the intuition that behaviors based on different classes of antidepressants should be more closely related to each other than to other classes of neuroactive drugs, such as stimulants. The dopamine $D_{2/3}$ agonizts lisuride and PD 128,907 also appeared in a similar region of space (labels 10,12). Antipsychotics clozapine and mianserin appeared closely in phenotypic space (labels 1,7). Through the lens of correlation distance instead (Supplementary Fig. 3), we see some similar high-level patterns but less behaviorome structure. For example, mianserin and loxapine are no longer neighbors; indeed, mianserin (label 1) appears closer to paroxetine (label 2)

than to loxapine (label 7). In other words, correlation distance places an antipsychotic closer to an SSRI than another antipsychotic, suggesting a lower clustering quality based on this region's canonical MOAs and indications.

## Model generalization to an orthogonal library of diverse drug-like compounds

We test the ability of the machine learning model to generalize to a library of diverse drug-like compounds from the DIVERSet, which had been screened months before the NT-650 set. The prior screen contained compounds with less than 0.3 Tanimoto Similarity, on average, to their closest match in NT-650 (Supplementary Fig. 14). Its goal was neuroactive compound discovery rather than quality control or model training; thus, it traded fewer replicates than NT-650 for greater compound diversity. We previously used a similar library to discover drug-like compounds that cause paradoxical excitation in larval zebrafish[10,28,29]. In that study, we performed the phenotypic screen with the DIVERSet library, and the resulting MI traces were compared (using correlation distance) against a reference drug, etomidate, that consistently induced a strong phenotype in the larval zebrafish.

Here, we investigate if the learned distance model outperforms correlation distance in identifying drug-like compounds from the DIVERSet library that cause similar phenotypes. Instead of focusing on a single known reference drug, we calculate a distance matrix for every

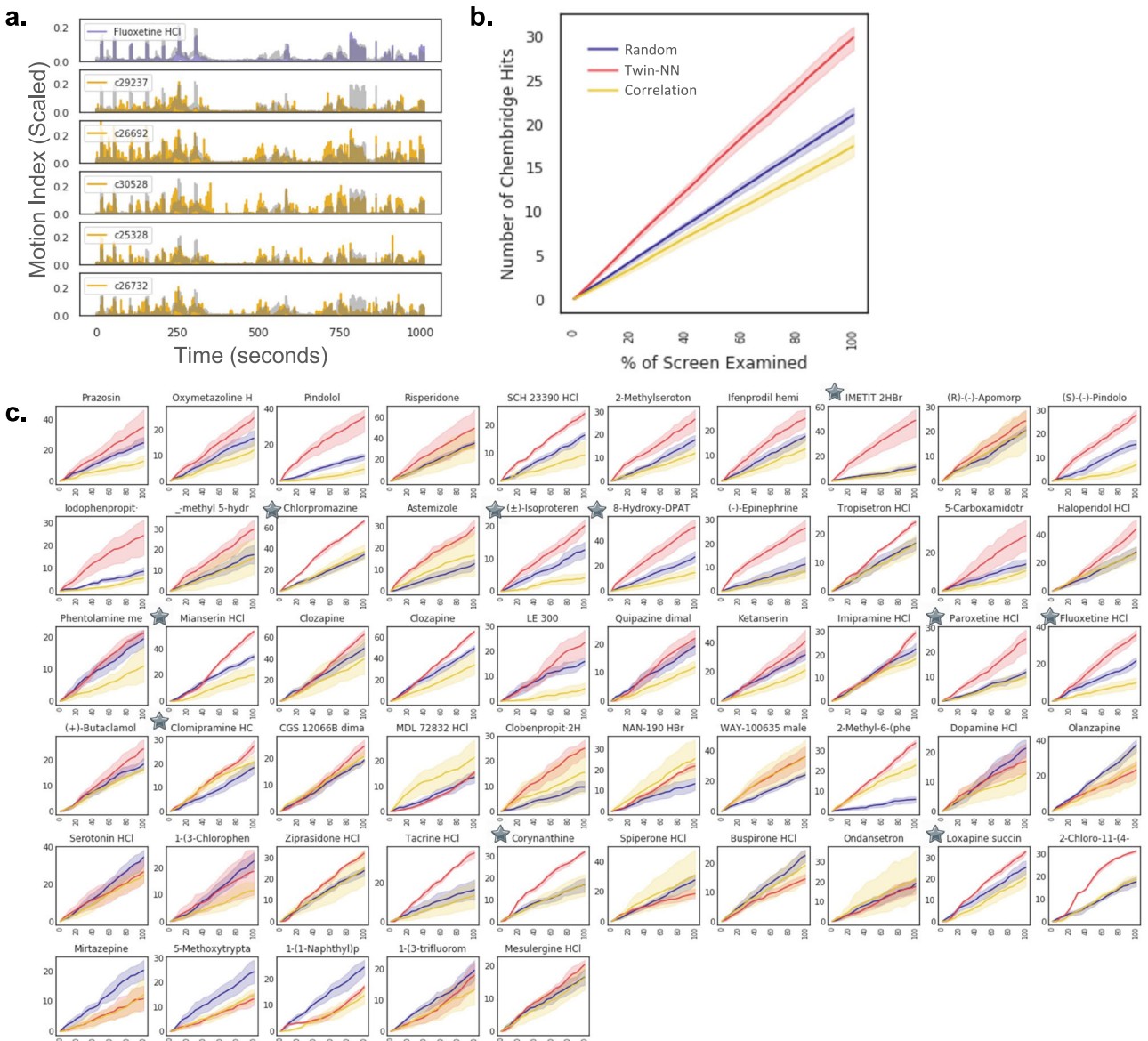

**Fig. 5 | Learned distance is a good proxy for the target bioactivity profile of DIVERSet compounds.** Assessed in a scaffold-agnostic screening paradigm, we compare motion index (MI) traces of NT-650 query compounds against a screened library of diverse compounds (Chembridge DIVERSet) using the Twin-NN learned distance and correlation distance versus a random baseline wherein the matched traces are randomly selected. **a** As an example, the fluoxetine MI trace (purple) from the NT-650 agrees well with the top 5 matched library compound traces (gold) ranked by Twin-NN distance. All time series in this plot are scaled to the minimum and maximum of the dataset (0 and 6750 MI units, respectively), and the y-axis is plotted on this normalized 0 to 1 scale. **b** We use a separate chemical informatics method, the Similarity Ensemble Approach (SEA[25,26]), to assess the library compound hits. Ranked by the similarity of their phenotypes to drugs from the NT-650 screen, we would expect that the likelihood of SEA target profiles between a query (NT-650) and its closest-match library (DIVERSet) compounds will increase with the

quality of the phenotypic distance metric. "Hits" (y-axis) are the number of DIVERSet compounds in a given sample that match their separate SEA profiles. "Sample" (x-axis) is the percentage of the DIVERSet library examined, where the analysis is limited to the top 500 matches from the library. The learned distance metric enriches for SEA hits better than correlation and the random baseline across the entire range of the screen. The plot uses a confidence interval of 95% via the seaborn lineplot function. **c** Similar to (**b**); but for specific NT-650 compounds selected by phenotypic strength (see "Methods"). Learned distance outperforms correlation and random distance, as with pindolol, imetit, and chlorpromazine. Correlation distance has significantly better enrichment for only one NT-650 compound, MDL 72832 (4th row, 4th column in grid plot). All subplots use a confidence interval of 95% via the seaborn lineplot function. Source data are provided in the Source Data File.

known compound in the NT-650 set against every compound in the DIVERSet library. Figure 5a shows an example of the top 5 compounds matching fluoxetine's phenotype. Evaluating performance in this context was challenging, as the diverse drug-like compounds lack known bioactivity ground-truth labels.

As one means of assessment, we use an established systems pharmacology tool, the similarity ensemble approach (SEA[25]), to predict MOAs for all DIVERSet compounds from their chemical

structures alone and compare these predictions against the established MOAs of NT-650 compounds that were their closest neighbors in the learned distance-metric space (see "Methods"). While SEA predictions are not perfect, they illuminate an otherwise dark MOA landscape of chemical matter. In a "phenosearch" approach, we rank-order and select the top 500 DIVERSet compounds by phenotypic distance to each known drug using correlation distance versus the Twin-NN models.

We observe a striking enrichment for known-target MOAs for the Twin-NN distance over correlation distance (Fig. 5b, d) based on the phenotypic associations for these diverse drug-like compounds. Twin-NN identifies more DIVERSet compounds with similar MoAs to the drug queries than correlation distance (Fig. 5b). Unexpectedly, random selection (as a null hypothesis; Fig. 5b, violet line) typically outperforms correlation distance at identifying shared MOAs, highlighting the limitations of correlation distance as a metric for time-series data such as MI. We also compare distance metrics by examining how often negative control wells match up with known drugs. With correlation distance, negative controls frequently rank in the top-500 "phenosearch" list for known drugs, but not by Twin-NN distance (Supplementary Fig. 4a, b). These findings suggest that the Twin-NNs are more effective than correlation distance at discovering DIVERSet compounds that induce similar phenotypes to known drugs and improve scaffold-hopping and neuroactive drug discovery for diverse drug-like chemical matter.

We also asked how the phenotypic space of the DIVERSet compares to the NT-650. We computed a combined UMAP (all compounds from NT-650 combined with DIVERSet), colored by the dataset the compounds came from (Supplementary Fig. 17). Strikingly, there is a large overlap between the blue (NT-650) and orange (DIVERSet) compounds, considering that the models were not trained on DIVERSet data. Despite this out-of-domain setting, on new compounds and behavioral data, the models usefully associate drug-like compounds from the DIVERSet with known compounds in the NT-650 set. This overlap supports the "phenosearch" approach, as most NT-650 query compounds have many phenotypically-similar DIVERSet compound neighbors in the map. Some regions have a higher density of NT-650 compounds and vice-versa, suggesting behaviors more commonly appear on one screen versus the other. Further studies might focus on selected sub-regions with DIVERSet density higher than NT650 to explore phenotypic space unexplored in the NT-650 phenotypic screen.

## Experimental validation of learned-distance metric in finding diverse drug-like compounds with shared pharmacology

Since the Twin NN models consistently enriched for predicted MOAs of DIVERSet compounds shared with known compounds (Fig. 5b), we sought to experimentally test these learned-distance predictions prospectively in a scaffold-hopping drug-discovery scenario. We selected 12 neuroactive drugs from diverse regions in the behaviorome UMAP (Fig. 4a). We purchased the top 5 DIVERSet compounds ranked by Twin-NN distance for each drug (60 compounds in total, of generally low Tanimoto similarity to their query drug; Supplementary Fig. 15). We hypothesized that the DIVERSet compounds acted through the same protein targets as those known for the drugs that the DIVERSet compounds mimicked phenotypically.

This was a straightforward logic in some cases: for example, IMETIT is a human Histamine $H_3$ agonist with activity at Histamine $H_4$[51]. We purchase and test the five DIVERSet compounds most closely ranked by the Twin-NN distance for direct binding to human Histamine $H_3$ and $H_4$, discovering binding of three of those compounds to $H_3$, at 1.1 μM, 0.99 μM, and 2.7 μM (binding affinity $K_i$), and one of them to $H_4$ at 5.4 μM ($K_i$) (Fig. 6c). In other cases, the choice of test targets was more complex, such as for the tricyclic antidepressant clomipramine, an inhibitor of serotonin and norepinephrine transporters with additional activity against other GPCRs, including serotonergic, dopaminergic, adrenergic, and histaminergic receptors. Furthermore, a compound's most potent activity in humans may not always account for its observed behavior in zebrafish. Off-target or side activities might cause the most pronounced response in the fish; this is an inherent limitation in the cross-organism study's design for polypharmacological drugs. Clomipramine's phenotypic location being closer to chlorpromazine than to the SSRIs fluoxetine and paroxetine

in the behaviorome (Fig. 4a), illustrates one such case. In humans, the clinical timescales involved in serotonin reuptake for behavioral modification are much longer[52] than the 1 h treatment duration used in our phenotypic screening, so we reasoned that the DIVERSet compounds phenomatched with fluoxetine might have acted through a subset of the targets it shares with chlorpromazine, such as serotonin 2B (5-$HT_{2B}$)[53,54]. Accordingly, two of clomipramine's top 5 phenomatched DIVERSet compounds achieve affinity ($K_i$) of 33 nM and 1.9 μM $K_i$s at 5-$HT_{2B}$ in prospective testing (Fig. 6c).

We test 216 new compound-target pairs based on 60 unique compounds and 17 unique protein targets. Of these, 8.3% are active at 10 μM or better Ki (Fig. 6a, b and Supplementary Table 2). IMETIT has the highest hit rate; 3 of its top 5 DIVERSet compounds have at least 50% inhibition at 10 μM or better against at least one of the targets; in the dose-response assays, two yields $K_i < 10$ μM, and the most potent, compound 58040, has a $K_i = 0.99$ μM for Histamine $H_3$. Overall, 7 of the 12 drug queries yield at least one DIVERSet hit for a 58% per-query hit rate; this corresponded to a 22% hit rate on a per-compound basis. All the dose-response binding curves from the secondary assays for the hits are provided (Supplementary Figs. 5–13). Where the tests failed, we may have picked the wrong subset of a query drug's protein targets to test against its DIVERSet compounds. For instance, clomipramine has known activities at a substantially wider range of targets than we could empirically test within the scope of this study, and this may account for mechanisms of action for those of its DIVERSet compounds that did not bind to 5-$HT_{2B}$.

## Learned phenotypic distances enable chemical scaffold hopping

Despite strikingly different chemical structures, the learned distance metric identified compounds that induced a similar behavioral phenotype in the case studies. We explored this idea further by comparing ECFP4[55] (chemical fingerprint Tanimoto distance) versus Twin-NN phenotypic distance for all possible combinations of two drugs from the randomized highly-replicated library used for training (Fig. 7). Here we define four quadrants: top-left (low Tanimoto distance, high Twin-NN distance), top-right (high Tanimoto and high Twin-NN), bottom-left (low Tanimoto and low Twin-NN), and bottom-right (high Tanimoto and low Twin-NN). Of potential interest in drug discovery efforts, the bottom-right region (dark purple) highlights where the commonly used cheminformatic means of comparing two molecules fail, but the Twin-NN distance succeeds.

Tanimoto chemical-structure distance does not correlate with phenotypic distance, except for isolated cases in the lower left (Fig. 7). Most pairs have Tanimoto chemical-structure distances greater than 0.4, despite sometimes inducing similar phenotypes through putatively shared MOAs. Dot size reflects observed MOA similarity, computed as a separate Tanimoto distance between the vectors of known-target activities for the two drugs derived from the ChEMBL 23 pharmacology database[51], "Methods"). The highest concentration of high-target-similarity compound pairs (large dots) favors regions where phenotypic distance is low, and chemical structure distance is average (0.3–0.7). This enrichment of known-MOA matches in the presence of good (low) Twin-NN phenotypic distance pairs is consistent with learned phenotypic distance predicting shared biological mechanisms. We hone in on these known-drug pairs with several thresholds (> 0.2 ChEMBL target-activity similarity, a chemical-structure distance > 0.5, and a Twin-NN phenotypic distance < 0.3), which yields 51 known-drug pairs that we rank by biological target similarity (full table provided in Supplementary Table 3). We note that there are two seemingly identical rows in the table for Clozapine with Chlorpromazine since there were two unique occurrences of Clozapine in the NT-650 plates provided by the supplier. The drugs in the top pair (7-OH-DPAT and ropinirole) are potent Dopamine $D_3$ agonizts and antiparkinsonian agents. Thus our Twin-NN phenotypic distance associates known

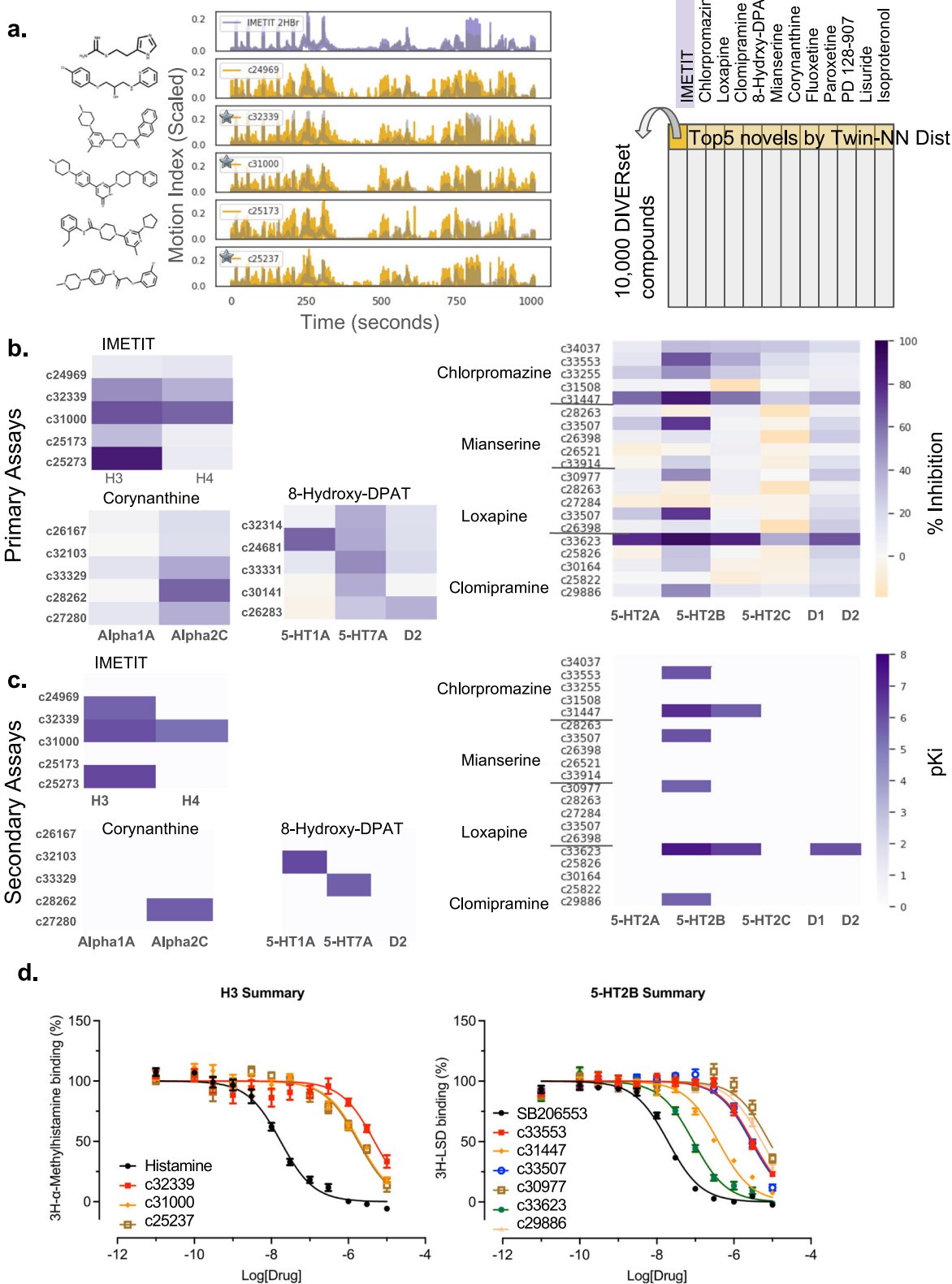

drugs with a shared mechanism of action but high chemical structure distance, highlighting its usefulness for scaffold hopping.

On the other hand, some pairs of drugs with high phenotypic similarity and middling structural similarity lack shared MOAs (small dots), which suggests these drugs induce similar phenotypic effects in larval zebrafish through different, parallel, or unstudied MOAs. These pairs correspond to a region of the known drug space of particular interest for drug discovery, and further studies might explore why

these pairs of known drugs are linked phenotypically in our study through potentially underexplored mechanisms.

## Discussion

Deep metric learning models trained on high-replicate phenotypic larval zebrafish screens identify pairs of drug-like compounds despite experimental variability, group human drugs based on zebrafish effect, find connections among compounds that traditional chemical data

**Fig. 6 | Prospective experimental validation against human receptors in vitro.** Using a cheminformatic protein target prediction approach (SEA[25,26]) with the Twin-NN phenotypic distance, we make the mechanism of action predictions for NT-650 compounds and test them experimentally by radioligand binding assays. **a** Left: Top 5 DIVERSet compounds (represented by their motion index time-series, rows 2–6, gold) matched by Twin-NN distance to the NT-650 drug Imetit (top row, purple). Right: Diagram of this "phenoblast" approach. NT-650 drugs are columns. DIVERSet compounds are rows, ordered by Twin-NN phenotypic distance. Supplemental Table 1 maps compound IDs to supplier IDs. All time series in this plot are scaled to the minimum and maximum of the dataset (0 and 6750 MI units, respectively), and the *y*-axis is plotted on this normalized 0 to 1 scale. **b** Primary radioligand binding assays (binding inhibition at 10 μM, %) for 7 known drugs. The heatmap shows 5 DIVERSet compounds selected for testing (rows), with the SEA-predicted human protein targets as columns. **c** Same as (**b**) for secondary assays (dose-response radioligand binding experiments). **d** Representative dose-response curves from (**c**) for selected DIVERSet compounds tested against two human targets: the histamine H3 receptor (left) and the 5-hydroxytryptamine 2B receptor (right). Results (mean ± SEM) from a minimum of 3 independent assays (each in triplicate) were normalized, pooled, and fitted to the built-in one-site competition binding function in the GraphPad Prism V10. Source data are provided in the Source Data File.

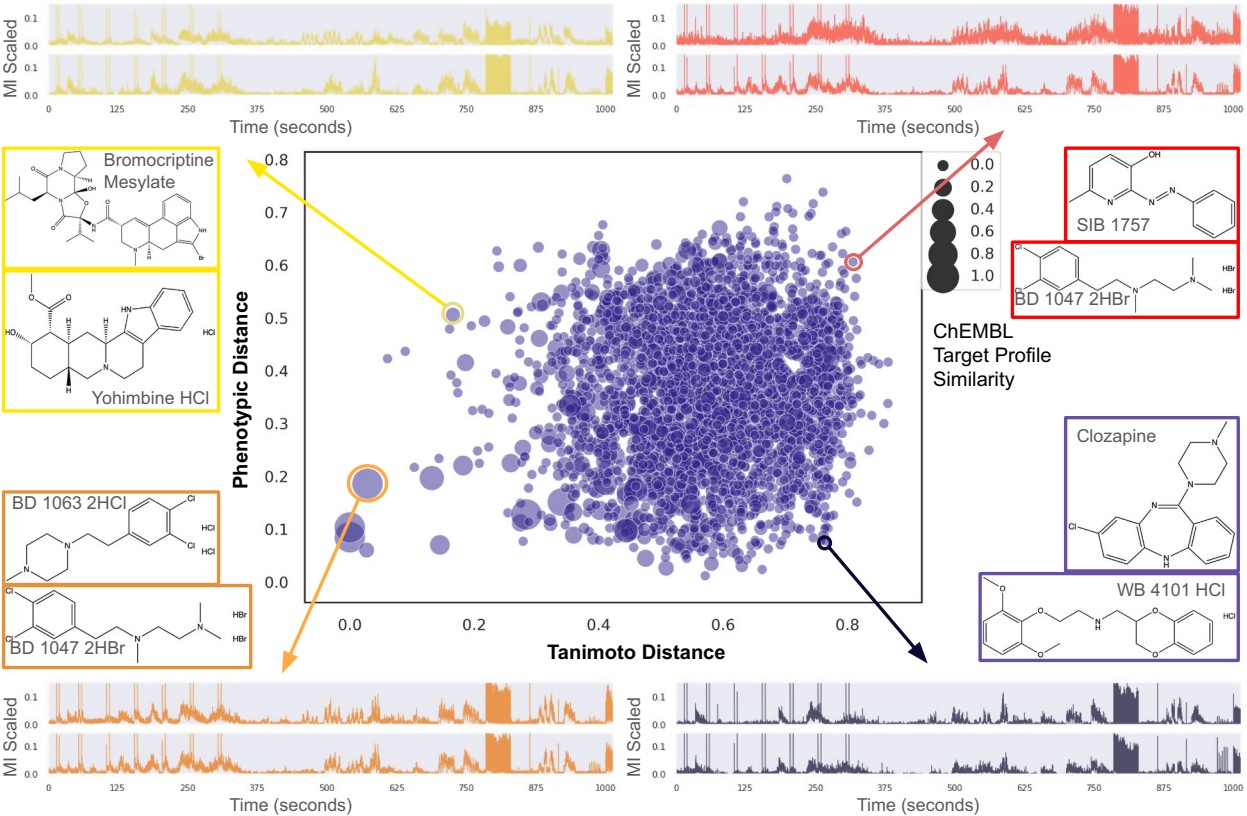

**Fig. 7 | Phenotypic screening with learned distance reveals scaffold hopping and drug prospecting opportunities.** Learned phenotypic distance (Twin-NN, *y*-axis) complements conventional chemical informatic distance based on chemical structure (Tanimoto coefficient on ECFP4 circular fingerprints, *x*-axis). The scatterplot contains all pairwise combinations of 83 NT-650 compounds (each pair is a dot), calculated from their average MI traces and chemical structures. Where the phenotypic distance is low (< 0.3) but the Tanimoto distance is average or high (> 0.4), molecular structure dissimilarity misses neuroactive similarity. We illustrate each quadrant with examples. Bottom left: low Tanimoto and phenotypic distance (both metrics agree that molecules are similar). Bottom right: low phenotypic distance and high Tanimoto distance (scaffold hopping opportunity). Top-left: low Tanimoto distance, but high phenotypic distance (classic "activity cliff": disparate activity despite high structural similarity). Top right: high Tanimoto and phenotypic distances (both metrics agree that molecules are unrelated). Dot size reflects the Tanimoto similarity between the target profiles (as binary vectors) of the compound pairs. All time series in this plot are scaled to the minimum and maximum of the dataset (0 and 6750 MI units, respectively), and the *y*-axis is plotted on this normalized 0 to 1 scale. Source data are provided in the Source Data File.

analyses fail to make, and group structurally distinct compounds by biological MOAs. These observations support using metric learning on large phenotypic screening datasets for drug discovery and scaffold hopping. Moreover, our first implementation of these complex learned-distance models fell prey to shortcut learning[41], wherein they exploited experimental artifacts in the screening dataset to achieve misleadingly high performance that did not generalize to similar but independent zebrafish screens. This deep *mis*-learning was nuanced and eluded conventional cross-validation, soundness checks, and exploratory data analysis tests. We believe the strategies described here to detect, correct, and stress-test the experimental screening datasets and revised models will find use in other studies that combine complex biological data with deep learning models.

Straightforward measurement methods like correlation, Euclidean, or dynamic time-warping distance fall short when identifying drugs whose replicates induced perceptible but subtle changes in zebrafish behavior (Fig. 3a, b). The main issue is that these methods cannot differentiate between irrelevant random variations and meaningful changes that illuminate the underlying pharmacology. Conventional metrics take all time points into account without weighting their importance. On the other hand, contrastive metric learning models disregard irrelevant parts of the data (features) and concentrate on the segments that display significant behavioral differences. For instance, clozapine- and DMSO-treated zebrafish exhibit periods of reduced motion (Fig. 1d, time points 600–700). Clozapine can look like a negative control by standard correlation methods,

which attribute equal importance to periods of inactivity and activity. In an extreme example, correlation distance scores two traces as almost identical when comparing a drug that sedates the fish except for a sudden movement spike versus a lethal control such as eugenol. However, a metric learning model learns that sudden motion spikes matter in differentiating drugs.

At a more global level, we construct a "behaviorome" - a visual map of drug similarity based on zebrafish behavior. This landscape, created by pairing zebrafish phenotype with an appropriate distance metric, reveals relationships between known neuroactive drugs and identifies underexplored areas with potential for drug discovery. From high-throughput behavioral screening data and the learned distance metric, we link human drugs directly to the in vivo vertebrate behaviors they induce. Classical informatic methods falter on diverse chemical structures, as they rely on the necessity of the similar property principle of chemical informatics[56]. This is particularly true at activity cliffs[57], where slight chemical structure changes drastically affect bioactivity. Phenotypic screening, using behavior, circumvents these limitations. Different compounds triggering similar zebrafish behaviors may interact with the same targets and pathways. The learned distance metric complements raw structural similarity (Fig. 7), underlining traditional cheminformatics limitations and opportunities for drug discovery and scaffold hopping.

We attempted to automate the discovery of diverse drug-like hits for disease-related mechanisms and pathways in the CNS. For new compounds, such as those from the Chembridge DIVERSet library, the Similarity Ensemble Approach[25,26,57] predicted unknown experimental MOAs. The metric learning models identified library compounds with marked enrichment in their predicted MOAs to the MOAs of known drugs, indicating pharmacological similarities (Fig. 5b, d). Instead of relying on in silico validation, we experimentally tested the predicted MOAs in vitro via prospective radio-ligand binding assays. We found that neuroactive drugs successfully linked to DIVERSet library compounds by phenotype and MOA 58% of the time. This hit rate surpassed early drug discovery hit rates using high throughput screening (HTS, 0.01–0.14%) or virtual screening (VS, 1–40%)[58]. Unlike typical HTS or VS hits, behavioral hits may offer more robust lead series starting points because they, by definition, already trigger an in vivo effect in zebrafish and show animal tolerance. Many in vitro hits fail in vivo due to absorption, distribution, metabolism, excretion (ADME) issues, and pharmacodynamic/kinetic properties such as blood-brain barrier penetration are crucial for neuroactive drugs. However, deep metric learning on behavioral screening data quickly identified hits that could circumvent these issues.

In an unintended but instructive project outcome, we grappled with the first metric learning models silently exploiting shortcut learning on the original dataset, which had not used randomized plate layouts. Despite passing conventional soundness check analyses, including label randomization and scrambling input features (y- and x-scrambling), the learning models exploited subtle experimental dataset artifacts. Pre-determined plate layouts from drug suppliers might inadvertently teach the models positional effects by exploiting slight irregularities in the experimental setup, such as minor differences in distance to directional light and sound sources (Fig. 1a). These effects, imprinted in high-frequency components of time-series traces, were imperceptible to humans but perceptible to deep learning models. This generalizability limitation was not an overfitting issue and could not be rectified by refining training-test set splits, such as scaffold-splitting drugs or time-series trace clustering. Consequently, we re-ran the full-scale experimental screen with robotically randomized plate layouts on the same compound library to assess this challenge unequivocally. Indeed, models trained on the original dataset deteriorated when we computationally smoothed high-frequency components of the motion index traces (Fig. 2b), but those trained on the fit-to-purpose randomized screen remained unaffected (Fig. 3b). While

we might instead have attempted to train generative adversarial networks (GANs)[59] to remove shortcut signals computationally[60], complex models such as GANs can be brittle, and we sought a more definitive analysis. As an intriguing challenge, follow-up studies by those interested in mitigating shortcut learning might find value in comparing new algorithmic versus the experimental plate-effect removal strategies on these two datasets.

We faced several practical caveats in the metric learning training procedures. Particularly, mismatched compound pairs within the same pharmacological class may trigger similar behaviors in zebrafish. We considered using Anatomical Therapeutic Chemical (ATC)[61] class or predicted protein target activity profiles by the Similarity Ensemble Approach (SEA)[25,26] to exclude misleading false-negative compound pairs from model training. However, many NT-650 substances are not approved drugs and lack ATC codes. Moreover, ATC classes operate across a hierarchy of varying branch depths, and it is likewise not clear what threshold to use for SEA-prediction similarity, given the ~2000 proteins in a target profile. Conversely, we might incorrectly label compound pairs as positive (false-positives) if they do not elicit a strong behavioral response. Inactive compounds could result from biological differences between humans and zebrafish, inactive concentrations, or limited effects on zebrafish behavior in our particular assay conditions. To tackle this, we deployed a separate random forest ("Methods") to remove inactive traces from positive-pair candidacy before metric learning training as a provisional solution. Consequently, the ground truth labels of compound phenotypic similarity used during model training are imperfect and noisy. Whereas improving these weak labels[62] may be an avenue for further refinement, we found them sufficient to train distance metrics robust to this biological label noise. Another caveat is that due to experimental scope, we could not account for compound dose, as all compounds were screened at a single concentration of 10 μM. However, models trained on such data could presumably infer phenotypes triggered by alternative doses of compounds, as they already distinguish DMSO wells from treatments. In the out-of-domain application of the models to the larger DIVERSet library, compounds populated the full gamut of behavioral space compared to compounds in the NT-650 training set, including those that were fully or partially inactive. In future directions, the models might serve as bridges to relate otherwise disparate perturbations, ranging from compound doses to functional genomics or the role of environmental changes.

Comparing our success rates with conventional single-target-based high throughput screening (HTS) or virtual screening (VS) presents different hurdles. Since we do not know the protein MOA for DIVERSet neuroactive compounds a priori, we tested these compounds against multiple predicted protein targets. Consequently, we calculate a best-of-hit rate, which provides more identification opportunities than a single-target screen. However, the lack of knowledge about which protein targets the DIVERSet compound impact makes direct comparisons with per-target success rates problematic. Our overall hit rate was 58%, which implies a 42% chance that a given query using a known drug would result in no protein-matched hits. These represent missed opportunities more than methodological failure. Here, errors in MOA prediction for DIVERSet compounds or cryptic but shared protein off-targets may cause unexpected associations between NT-650 and DIVERSet compounds. However, as cheminformatic target prediction accuracy improves, this will further complement the phenotype-based metric learning approach. Finally, we acknowledge that the larval zebrafish animal model for studying neuroactive drugs has limitations due to genetic, anatomical, and behavioral complexity differences with mammals. Consequently, we must carefully vet the compounds, pathways, or behavioral phenomena identified in the larval zebrafish in more advanced animal models and humans to establish their therapeutic import, which is beyond our scope. This study's blend of screening technology and metric learning

is thus a tool to complement but not replace accepted animal models and methods.

Deep metric learning models with high-replicate behavioral zebrafish screens directly reveal scaffold hopping opportunities. These models outperform traditional distance metrics and cheminformatics methods, accurately classifying and grouping compounds from their zebrafish behavior alone. Prospective testing confirmed most of the predicted neuroactive MOAs using human receptors in vitro. Deep metric learning enriches phenotypic screening, yielding diverse drug-like compounds with actionable neuroactive effects despite different chemical structures. Despite the challenges presented by experimental variability and shortcut learning—where models exploited experimental artifacts—we successfully redeployed the screen and stress-tested the models, creating a robust approach applicable to diverse investigations that pair complex biological data with deep learning. Closely integrating fit-to-purpose larval zebrafish behavioral screening with deep metric learning is an efficient and robust way to identify new neuroactive compounds in vertebrates.

## Methods

### Ethics statement
All zebrafish procedures were performed and approved according to UCSF's Institutional Animal Care Use Committee (IACUC) and the Guide to Care and Use of Laboratory Animals (National Institutes of Health 1996) and conducted according to established protocols that complied with ethical regulations for animal testing and research.

### Animal husbandry
Eggs from a wild-type Singapore strain were collected by group matings and raised on a 14/10-hour light/dark cycle at 28 °C in egg water (GCULA) until 7 dpf. 8 healthy larvae were then distributed by pipette into the wells of 96-well plates. They were then incubated pre-treatment for 1 hr, dosed, incubated post-treatment for 1 hr, and then screened in the behavioral instrument. The zebrafish larvae used in these studies have not undergone sexual differentiation at this stage, eliminating sexual dimorphism as a potential confounding factor. In addition, the larvae are sourced from large group spawnings of wild-type zebrafish, which ensures a genetically diverse and balanced mix of male and female progenitors, further supporting the generalizability of the findings.

### Chemical libraries
Two chemical libraries were used in our study: the SCREEN-WELL Neurotransmitter Set (Enzo Life Sciences, Farmingdale, USA https://www.enzolifesciences.com/BML-2810/screen-well-neurotransmitter-library-10-plate-set/ and the ChemBridge DIVERSet Screening Library https://www.chembridge.com/screening_libraries/diversity_libraries/. Additional information about the chemical libraries, such as all compound structures (in SMILES format), is provided in Supplementary Data 1.

### Screening platform
The screening platform is described in detail[7] (Fig. 1 and "Methods"). The QC set screening methods are also described in that study. For the randomized experiments using the Screen-Well Neurotransmitter Set, we randomized the plate layouts with a custom code provided with this study. We transformed the physical layout of the plates accordingly using a BioMek robot in the Arkin lab at UCSF. For the NT650 screen, we treat each plate with 8 DMSO controls, 2-3 H2O controls, and 2-3 Eugenol (toxic) controls. Throughout the study we treat DMSO and H2O controls identically since we observe no significant differences in the motion index time series between these two types of negative controls. For the DIVERSet, only DMSO controls are included. Additional details are provided in the Supplement (Supplementary Tables 4, 5).

### Data collection
Larval zebrafish are plated onto 96-well plates (8 fish per well), and wells are dosed with drugs at 10 μM for primary screening (Fig. 1). Various stimuli such as acoustic sounds and physical tapping of the plate platforms are performed to elicit diverse behavioral responses in the fish, as optimized by Myers-Turnbull et al.[7]. Videos are recorded of the fish behavior for the duration of the experiment, which for the screens discussed in this work is around 14 min. For each well, the videos are encoded and converted into bulk motion over time, resulting in a one-dimensional time series or motion index (MI). Specifically, we used pre-interpolation m' values[7] defined by:

$$m'(I^t) = \sum_{ij} \mathbb{1} \left| I^t_{ij} - I^{t-1}_{ij} \geq 10 \right| \tag{1}$$

where $I^t$ is the grayscale image matrix at 1-indexed frame t. These videos represent the average motion across all 8 fish in each well. Since zebrafish movement can be uncoordinated, averaging over multiple fish can greatly improve the signal-to-noise ratio for many classes of drugs[7,10].

### Data analysis
We performed data analysis using scripts in GraphPad Prism v10 and Python (e.g., numpy (1.16.3), pandas (0.24.2), scipy (1.5.1), sklearn (0.20.3), and seaborn (0.11.1), rdkit (2019.03.2), pytorch (1.1.0)). SEA (Python code) is referenced in Keiser MJ et al. *Nat Biotech*, 2007[25]. For further details, see the study code repository (Code Availability).

### Filtering ineffective and lethal compounds by Random Forest
For the first high-replicate screen, we trained a random forest classifier to identify ineffective compounds (mimicking DMSO) using sci-kit learn[63]. The inputs are the MI of drugs and DMSO-treated wells, and the output is a binary label (effective or ineffective). We first split the entire dataset using an 80/20 train/test split. There were fewer DMSO-treated examples, so we randomly undersampled from the drug-treated wells to match the number of DMSO-treated wells. This resulted in 556 examples from each class in the training set and 180 examples in each class in the test set. For the randomized high-replicate screen, we include positive controls (eugenol), which is lethal to the larval zebrafish. Here, the random forest is trained to label MI into one of 3 possible bins (effective, ineffective, or toxic). As before, we first split the entire dataset using an 80/20 train/test split. There were fewer toxic examples than in the other 2 classes, so we randomly undersampled from those classes to match the number of toxic examples. This resulted in 100 examples for each class in the training set and 28 examples for each class in the test set (using an 80/20 train/test split).

### Conventional metrics used to calculate phenotypic distance
We used the sci-py package[49] to compute correlation, euclidean, and the fast-dtw[64] python library to compute the dynamic time warping distance between the 101,250 frame-long MI time series traces.

### Classifying quality control drugs with distance metrics
We trained a kNN (k-Nearest-Neighbors)[65] algorithm as implemented in the sklearn KNeighborsClassifier package to classify the quality control (QC) 16 neuroactive drugs based on their motion-index time-series traces using the implementation in the scikit-learn package[49,63]. Each QC compound was screened in replicates of 10, so we split the dataset into train and validation splits (8 train, 2 validation) replicates for each drug. The task of the KNN was to predict, for a given time series, which one of the 16 drugs it most closely corresponds to. We chose 15 for the number of nearest neighbors parameter.

### Training deep metric learning models for phenotypic distance
Our Twin-NN and Twin-DN models use a Contrastive Divergence loss function and may be thought of as contrastive learning approaches

that simply use real-world (experimentally collected) compound-replicate data instead of relying on the synthetically generated data-augmentation procedures used in contrastive learning architectures such as SimCLR[66]. Thus, our models are formally metric learning models because we provide a training label for each compound pair, reflecting whether the MI trace replicates are of the same small-molecule compound. Conceptually, however, the models are motivated by contrastive learning because they rely on a contrastive logic on replicated observations, albeit without requiring post hoc synthetic augmentations. Thus, a formal contrastive learning model like SimCLR that learns from imperfect synthetic augmentations would be unlikely to perform better than our approach that directly leverages the real-world compound replicate readouts we purposefully collected in a fit-to-purpose way specifically for this study.

In general, a dataset of positive and negative pairs is required to train a Twin-NN or Twin-DN model. To construct such a dataset, we screened the SCREEN-WELL Neurotransmitter Set ("NT-650") in high replicate. We define any two replicates of the same compound to be positive pairs, while a replicate of one compound paired with a replicate of another compound was a negative pair. Each plate of 96 drugs was replicated 7–10 times, creating a sizable dataset of positive and negative pairs. In practice, not all compounds will exhibit an observable effect in larval zebrafish, so it can be misleading to label replicates of such drugs as "positive" (see Discussion). Thus, we filtered all pairs where at least one pair member was "ineffective" or "lethal" by the random forest. We trained the second round of Twin-NNs similarly but using the fully randomized dataset instead.

To create the dataset of pairs, we first split the dataset by drug to minimize memorization and over-fitting, allocating 80% of drugs for training and 20% for the test set. This scheme naturally presents a generalizability challenge for the models, since phenotypes induced by the training drugs might not be induced in the test drugs. Data splitting encourages the models to learn fundamental features that are independent of drug or phenotype, which can lead to much better generalizability. Next, we performed a class balancing procedure. There are many more pairs that can be enumerated across different drugs (negative pairs) than pairs from the same drug (positive pairs), but this could lead to class imbalance issues during training. Hence, we randomly subsampled the negative pairs to match the number of positive pairs for both the training and test sets. Next, we aimed to include additional commonly encountered pair types. For positive pairs, we often encountered control-control and tox-tox pairs, and for the negative pairs, we often encountered control-drug, control-tox, and tox-drug pairs. To ensure the models were exposed to enough of these pairs, we ensured that 25% of the total positive pairs in the dataset came from the control-control or tox-tox classes (allocated evenly) and that 25% of the total negative pairs in the dataset came from the control-drug, control-tox, or tox-drug classes (allocated evenly across these 3 classes). We saved pairs to NumPy arrays of indices that corresponded to indices of the time series data and provided both the pair arrays and raw data in our online data repository.

We used PyTorch to train the Twin-NN and Twin-DN models (see Github repository). Briefly, we loaded the positive and negative pairs using a Pytorch Dataloader, randomly swapped the pair order, sampled at every 5th frame, and min-max normalized them. Then we passed these pairs of motion index time series through the MLP architecture (Twin-NN) or Dense architecture (Twin-DN). The Twin architecture was a 6-layer feed-forward neural network. The input layer size was 20250 (length of the input); each subsequent layer was 4000, 500, 250, 100, and 10, respectively. After each linear layer (except the last), we performed batch normalization and ReLU activation. We passed each time series from an input pair in this feed-forward architecture, after which we computed a contrastive loss from the 2 outputs (vectors of length 10 each). We used a margin of 0.5 for the negative pairs in the contrastive loss. We back-propagated the contrastive loss and updated model weights after each batch, until reaching convergence or the maximum epoch count. We used the same training procedure for the Twin-DN models, except that we based the architecture on DenseNets instead. The final output for each input from the DenseNet was also a vector of length 10. To train the models, we used a learning rate of 5e-4 with the Adam optimizer and weight decay set to 1e-6. For the Twin-NN model, we used a batch size of 32. For the Twin-DN model, we used a batch size of 8, as this was the largest batch size to fit in GPU memory. We used an NVIDIA GeForce GTX 1080 Ti GPU on a CentOS Linux kernel 3.10.0 operating system with an x86-64 architecture.

## Behaviorome UMAPs

To plot a UMAP embedding for the NT-650 dataset (Fig. 4), we computed the mean time series for every compound in the dataset (across all replicates of the compound). We then calculated an NxN distance matrix based on these mean times series. This distance matrix is passed into the UMAP library from Python[47]min_dist parameters to 10 (default is 15) and 0.1 (same as default), respectively.

For the behaviorome with supplier-provided class labels (Supplementary Fig. 16), we performed the analysis for the 325 compounds with strong predicted phenotypes ("Methods"). We computed the UMAP embedding using the same procedure as for Fig. 4, except that points are colored by the class labels, keeping the n_neighbors parameter the same and setting min_dist to 0.25. For the joint UMAP between NT-650 and DIVERSet (Supplementary Fig. S17), we applied the same procedure to the combination of the two datasets (all unique replicates of NT-650 and all unique wells of DIVERSet), but setting the n_neighbors parameter to 50 while using the default value for min_dist (0.1). For compounds highlighted in the legend, we computed the mean embedding point across all replicates of the compound.

## Stress-testing the metric learning models

We performed high-frequency signal filtering using a Hanning smoothing filter[48] as implemented in the scipy package[49], using a window size of 11. We tried three adversarial controls on our Twin-NN models: label-shuffling, input randomization, and predicting well distance (Supplementary Fig. 2). For label-shuffling, we randomly shuffled the labels while keeping the input fixed. For input randomization, we generated random MI vectors of the original length using the Python NumPy package[67], while holding the labels fixed. For well distance, we used the Twin-NN models to predict well distance (computed as the Euclidean distance between pairs of wells). We define as "same" those pairs that are neighbors below a certain distance cutoff (2 and 5.2 plate distance units), and as "different" those pairs that are distant from each other (above the chosen cutoffs). The Twin-NN models appear to have no ability to distinguish between pairs in this context at either of these cutoffs, suggesting a lack of strong signal in the motion index time series attributable to plate location alone.

## Chemical informatics and bioactivity prediction

Chemical structure similarity is computed from ECFP4 fingerprints and Tanimoto similarity using the rdkit package. Bioactivities of compounds of unknown indication or mechanism of action (as for hits from the SCREEN-WELL Neurotransmitter Set) are computed using the similarity ensemble approach (SEA[25,26]) together with version 23 of ChEMBL[51]. We use a SEA p-value cutoff of 1e-25 for bioactivities.

## In vitro binding assays against human receptors

The 60 compounds from the DIVERSet screen were tested in radioligand binding assays (performed as previously described[68–70]) against human receptors at the National Institute of Mental Health Psychoactive Drug Screening Program at UNC (PDSP). Primary inhibition screens were performed at the final dose of 10 μM in the in-plate quadruplicate set, and compounds passing a threshold of 50% inhibition were subjected to secondary dose-response assays to determine

binding affinity (Ki) in the intra-plate triplicate set. All binding assays are conducted in 96-well plates at the final volume of 125 μl per well. In brief, samples, radioligands, and receptor membranes are mixed and incubated for 60 min at room temperature in the dark. The reactions are terminated by rapid vacuum filtration onto glass fiber filters chilled with 0.3% PEI solution. The filters are quickly washed twice with cold wash buffer (50 mM Tris HCl, pH 7.4) and dried. Radioactivity is read in a Microbeta counter. Results are analyzed in GraphPad Prism V10. Detailed assay protocols and procedures are also available from the NIMH PDSP homepage (https://pdsp.unc.edu/pdspweb/?site=assays).

## Use of Large Language Models (LLMs)

We used OpenAI ChatGPT 3.5 Turbo and ChatGPT 4 as scientific editing tools when writing the manuscript. We prompted the LLMs to suggest revisions to our manually drafted text for improved clarity and conciseness at the sentence and paragraph levels. We did not ask the LLMs to generate content de novo. An example of a prompt we used was, "You are helping edit papers for a broad scientific audience, emphasizing clarity and conciseness. Revise the following paragraph: < draft text here >." We manually reviewed the LLMs' suggested revised text word-by-word and decided whether to include parts, all, or none.

## Reporting summary

Further information on research design is available in the Nature Portfolio Reporting Summary linked to this article.

## Data availability

The processed motion index time series data used in this study is available at: https://zenodo.org/records/10652682 (https://doi.org/10.5281/zenodo.10652682). Source data are provided with this paper as Source Data file. Source data are provided with this paper.

## Code availability

The source code and model weights underlying the Twin-NN and Twin-DN models are available at: https://github.com/keiserlab/deepfish (https://doi.org/10.5281/zenodo.13910211)[71].

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

## Acknowledgements

This work was supported by grant DAF2018-191905 (https://doi.org/10.37921/550142lkcjzw) from the Chan Zuckerberg Initiative DAF, an advised fund of the Silicon Valley Community Foundation (funder https://doi.org/10.13039/100014989) (M.J.K.), the Paul G. Allen Family Foundation (D.K. and M.J.K.), and the US National Institutes of Health (NIH) grant R01AA022583 (D.K.). Binding results (primary and secondary $K_i$ determinations) were generously provided by the National Institute of Mental Health's Psychoactive Drug Screening Program, Contract # HHSN-271-2018-00023-C (NIMH PDSP). The NIMH PDSP is Directed by Bryan L. Roth, MD, PhD at the University of North Carolina at Chapel Hill and Project Officer Jamie Driscoll at NIMH, Bethesda, MD, USA.

## Author contributions

Conceptualization: L.G. and M.J.K.; Methodology: L.G. and M.J.K.; Software: L.G. and D.M.T.; Validation: L.G. and M.J.K.; Formal Analysis: L.G. and D.M.T.; Investigation: J.T., D.M.T., M.N.M., and S.C.; Resources: D.K.

and M.R.A.; Data Curation: D.M.T.; Writing - Original Draft: L.G.; Writing - Reviewing and Editing: L.G. and D.M.T., and M.J.K., Visualization: L.G.; Supervision: M.J.K., D.K., and M.R.A. Project Administration: L.G. and M.J.K.; Funding Acquisition: D.K., and M.J.K.

## Competing interests

The authors declare no competing interests.
