## [Transparent Peer Review file · Nature Communications]

Deep phenotypic profiling of neuroactive drugs in larval zebrafish

Corresponding Author: Professor Michael Keiser

This manuscript has been previously reviewed at another journal. This document only contains reviewer comments, rebuttal and decision letters for versions considered at Nature Communications.

Version 0:

Reviewer comments:

Reviewer #1

(Remarks to the Author)

In this manuscript, authors utilized a library of 650 ligands to investigate their effects on larval zebrafish behavior through high-replicate screening. They employed twin neural networks (NNs) for drug phenotyping, but encountered challenges due to "shortcut learning" phenomena, wherein models exploited unintended dataset artifacts. To address this, they retrained deep metric learning models on synthetically randomized datasets to mitigate confounding effects, prompting a redesign and re-collection of experimental screens. Authors claimed that the revised models could cluster neuroactive compounds and facilitate scaffold hopping, demonstrating a strong correspondence with known neuroactive biology. Additionally, authors claimed that the learned distance metric exhibited generalization capabilities to novel compounds, enabling automated discovery of neuroactive compounds active on human receptors during prospective *in vitro* testing.

The main contribution of the manuscript is to apply an off-the-shelf neural network model to predict the phenotype similarity of zebrafishes. The effectiveness of the proposed models remain questionable due to the drawbacks in the experimental design. Specifically,

Major:

1. The training and testing procedure used in the manuscript is not the common practice in the deep learning. The performance can be over-estimated. The data should be split into training, validation, and testing. The hyperparameters should be determined by the validation set, but not the testing data.
2. To evaluate if the proposed twin-DN and twin-NN is generalizable, the chemicals in the testing data should be significantly different from those in the training and validation data, e.g., different MOA.
3. Page 8, line 204, and Figure 4, UMAP will not generate principal components. Please explain.
4. Please provide more details on "novel library" from the DIVERSet (Page 9, line 220). What are the chemical structural similarities between the novel library and the NT-650 set? Similarly, the structural similarity between the selected drugs for experimental testing and the reference molecule should be reported.
5. The successful rate of experimental validation is moderate. It does not provide adequate support for the effectiveness of proposed methods.

Minor:

1. Page 6, line 150, "Twin-SN" is "Twin-NN"?
2. It will be interesting to see if contrastive learning will perform better than Siamese network.

Reviewer #2

(Remarks to the Author)

In this manuscript, Gendelev et al. present a powerful machine learning-based approach to identify and characterize neuroactive drugs. The authors first generated time series of aggregate movement across multiple zebrafish larvae that had been exposed to a sequence of sensory stimuli in the presence or absence of selected small molecules. They then developed a pipeline based on twin neural networks to distinguish specific behavioral changes from normal behavioral profiles. The authors found that the new machine learning-based pipeline performed much better than previously established methods, and the trained models clustered neuroactive compounds in good agreement with their known mechanism of action. They also identified several new compounds that elicited phenotypes similar to previously characterized neuroactive compounds. In vitro data with human targets acquired by an external screening service confirmed many of the new hits and identified compounds with low micromolar binding affinities to their predicted targets, which is a major success of this work. A clever pairwise comparison between the behavioral phenotypes and corresponding differences in the chemical structures of the drugs is a significant addition to the field. The authors round off the paper with a well-balanced discussion of the limitations of their work. In addition to the software code, the data sets are available from a Zenodo repository as an extra resource for the community. As detailed below, a few minor points should be addressed before publication to improve the clarity of presentation for a broad readership.

- 1) It would be useful to highlight the stimuli that were applied in Fig. 1d, so that readers can interpret the MI changes.
- 2) Please add a y axis definition and scale for figures such as Fig. 1d, Fig. S1 etc.
- 3) Please describe in the figure legend what the orange color in Fig. 1e indicates.
- 4) It would be helpful to improve the clarity of Fig. 3d. As the y axis indicates "Cumulative #Drugs", it seems that the integral of the curves should be the total number of drugs. The integral is expected to be identical for Twin-NN and Corr, but the areas under the curves differ. Please clarify.
- 5) In the Results section, please add the data for "not shown" or change the statement.
- 6) Some figure references seem to be incorrect in the text (e.g. Fig. 3a-b in the second paragraph of the Results section).
- 7) It would be useful if the primary data for the results in Fig. 6b could be shown in the Supplementary Material.
- 8) The description "We color the dots ..." does not seem to agree with the presentation in Fig. 7. Please clarify.
- 9) Table T3 contains duplicates (e.g. Clozapine – Chlorpromazine HCl). Please clarify.
- 10) The statement "Individual data points are shown when possible, and always for $n \leq 10$ " in the Editorial Policy Checklist does not seem to be correct for Fig. S5-13.
- 11) References to other recent papers that have used twin neural networks in zebrafish studies (e.g. PMIDs 35309679, 37996754) could be included.

Reviewer #3

(Remarks to the Author)

The manuscript by Gendelev et al. entitled "Deep phenotypic profiling of neuroactive drugs in larval zebrafish" describes a deep learning-assisted approach for characterizing and classifying behavioral changes induced by neuroactive drugs in larval zebrafish. Although the screening approach has been previously reported by the authors, the analysis approach by contrastive learning is novel and significantly outperforms previous correlation distance-based methods. In addition to establishing a drug-induced zebrafish "behaviorome", the authors discovered novel druglike compounds with diverse structure at a staggering hit rate of 58% through an additional screen of uncharacterized compounds followed by validations via in vitro binding assays. The characterization of a shortcut learning phenomenon is also insightful. The study represents a significantly advance in phenotypic screening methodology and is suitable for publication in Nature Communications with the following suggested edits.

- 1: Figure 4: The behaviorome is apparently classified into distinct clusters. Are different clusters enriched in neuroactive drugs belonging to different functional categories? If so, please annotate.
- 2: I would be very interested to see the DIVERset behavioral profiles clustered with the NT-650 dataset in the same UMAP plot as a supplementary figure. I'm wondering if you were able to expand the behaviorome established by the NT650 compounds through behaviors generated by the DIVERset compounds. Were you able to discover behavioral profiles that significantly deviated from the NT650 compounds?
- 3: The same drug can elicit different behavior at different doses, and the correlation between dose and behavior changes

might not be linear. I understand the decision to focus on one dose for the initial platform setup, but would be very interested to see how this model generalizes to different doses of the same drug, such as by testing different doses of a few selected drugs and see how they rank by Twin-NN distance among other drugs screened. I would also be interested to see a discussion of whether future updates of the model can be trained with different drug doses.

4: The paper PMID:36814839 describing another deep learning-assisted behavioral classification method in zebrafish is relevant and should be included in the reference.

Version 1:

Reviewer comments:

Reviewer #1

(Remarks to the Author)

The major concerns have been clarified or addressed. Thanks.

I have also been asked to comment on the concerns of reviewer #2.

Reviewer 2's comments make sense. Authors have addressed all points raised by Reviewer 2.

Reviewer #3

(Remarks to the Author)

The authors have thoroughly addressed all my previous comments. I have no further comments and recommend the manuscript to be accepted for publication.

Response to Reviewer Comments

We thank the Reviewers for their support of the work and their thoughtful and actionable feedback, nearly all of which we have adopted and implemented. This includes substantive additions or clarifications to the text, new scientific analyses, revised Figures, and four new Supplementary Figures, which have improved the study.

Reviewer 1

In this manuscript, authors utilized a library of 650 ligands to investigate their effects on larval zebrafish behavior through high-replicate screening. They employed twin neural networks (NNs) for drug phenotyping, but encountered challenges due to "shortcut learning" phenomena, wherein models exploited unintended dataset artifacts. To address this, they retrained deep metric learning models on synthetically randomized datasets to mitigate confounding effects, prompting a redesign and re-collection of experimental screens. Authors claimed that the revised models could cluster neuroactive compounds and facilitate scaffold hopping, demonstrating a strong correspondence with known neuroactive biology. Additionally, authors claimed that the learned distance metric exhibited generalization capabilities to novel compounds, enabling automated discovery of neuroactive compounds active on human receptors during prospective in vitro testing.

The main contribution of the manuscript is to apply an off-the-shelf neural network model to predict the phenotype similarity of zebrafishes. The effectiveness of the proposed models remain questionable due to the drawbacks in the experimental design. Specifically,

We thank the Reviewer for their detailed constructive feedback and suggestions, which demonstrated a close read of the study and were well-founded. We hope the clarifications, text modifications, and new analyses introduced below will address the Reviewer's specific concerns.

We also wanted to address a few apparent misapprehensions within the short summary above that may otherwise have inadvertently minimized the study's perceived impact:

1. *New high-replicate behavioral screens in vertebrates*

We did not utilize an [existing] library of 650 ligands. Specific to this study, several of the authors designed, prepared, performed, and recorded a behavioral screen of 7-10 well replicates for each of 650 compounds, comprising approximately 5,525 compound-induced behavioral traces on more than 44,000 larval zebrafish. Each MI trace contains 101,250 time points. Then we did it a second time, incorporating physical robotic well randomization. This yielded approximately 1.1 billion collected time points on 88,000 compound-treated fish, not accounting for the negative control wells— a considerable undertaking. We release these unique before-and-after compound-induced behavioral trace datasets for open research and discovery use.

2. *Physical randomization guided by wet-dry lab integration*

We did not only "synthetically randomize" the datasets but also ultimately performed full physical randomization by repeating the entire screen. This was a significant effort in collaboration with the UCSF Arkin Lab on a BioMek liquid-handling robot. Synthetic randomization alone could not unequivocally demonstrate shortcut learning, especially when the shortcuts were unknown.

For clarity, we have added the word "physical" to the abstract: "The machine learning initially exploited subtle artifacts in the phenotypic screen, necessitating a complete experimental re-run with rigorous physical well-wise randomization."

We likewise specify physical randomization in the Methods: "For the randomized experiments using the Screen-Well Neurotransmitter Set, we randomized the plate layouts with a custom code provided with this study. We transformed the physical layout of the plates accordingly using a BioMek robot in the Arkin lab at UCSF." We also state in the Results: "To unequivocally control for within-plate positional confounding effects, we perform a second high replicate screen of NT-650, but this time with the treatments fully robotically randomized across plates

and wells (Methods).” We have further emphasized several mentions of physical randomization, as in the Introduction, as well.

3. Study-specific neural network models

We did not employ off-the-shelf neural network models (e.g., as one might find at <https://huggingface.co>). Instead, we started with a PyTorch Siamese network open source codebase to create this study's Twin-NN and Twin-DN zebrafish models several years ago. The Twin-DN architecture for instance employs a nonstandard reworking of DenseNet blocks, which were of course originally designed for 2D (image) inputs rather than 1D (time series) MI traces.

Major:

1. The training and testing procedure used in the manuscript is not the common practice in the deep learning. The performance can be over-estimated. The data should be split into training, validation, and testing. The hyperparameters should be determined by the validation set, but not the testing data.

We wholeheartedly agree with the Reviewer about the value of a strong hold-out set. We regret if it was not clear in our presentation of the materials, but we use a strictly held-out set on which we performed blind predictions subjected to prospective wet-lab testing as our real-world assessment of model generalizability. Computationally, we split the NT-650 dataset (325 compounds after filtering to only those with predicted strong phenotypes, per Methods) into training (80%, n=260 unique compounds) and validation (20%, n=65) sets, stratified by compound—a more challenging split than the usual random, with no replicates in common between the training and validation splits (see 3rd paragraph of the section, “Training deep metric learning models for phenotypic distance” in Methods). Nonetheless, as the Reviewer points out, even difficult train-validation splits may mask failures in model generalizability. To assess this, we used a large separate dataset, experimentally collected at a different time, from a notably different compound library (DIVERSet, n=10,000 compounds) as the strict hold-out set for the prospective in vitro testing. This diverse and out-of-distribution testing compound library was structurally distinct from the train+validation compound library, as illustrated in the new Supplementary Fig S14. All hyperparameter optimization was restricted solely to the initial NT-650 dataset.

2. To evaluate if the proposed twin-DN and twin-NN is generalizable, the chemicals in the testing data should be significantly different from those in the training and validation data, e.g., different MOA.

We agree. Indeed, the testing (DIVERSet) and train+validation (NT-650) compound sets are significantly different, with their peak pairwise molecule similarity distribution between matched compound pairs ≤ 0.3 Tc, as shown in the new Supplementary Figure S14. Furthermore, the testing dataset is much larger than the train-validation dataset (as noted in point 1 above), making it likely that the especially-diverse testing dataset would be more likely to contain compounds with properties outside those in the train+validation dataset than the other way around.

3. Page 8, line 204, and Figure 4, UMAP will not generate principal components. Please explain.

Good catch! We’ve corrected the axis labels.

4. Please provide more details on “novel library” from the DIVERSet (Page 9, line 220). What are the chemical structural similarities between the novel library and the NT-650 set? Similarly, the structural similarity between the selected drugs for experimental testing and the reference molecule should be reported.

We were happy to add these new analyses. As reported in response to points 2 and 4 above, Supplementary Figure S14 shows the distribution of all-by-all pairwise chemical structural similarity between the novel library (testing; DIVERSet) and the NT-650 (training+validation) set, which peaks at no more than 0.3 Tanimoto Similarity. We have also generated a new Supplementary Fig. S15 to quantify the similarity between the selected prospectively tested compounds and their reference molecules. The chemical structural similarities between the NT-650 drugs (reference) and novel DIVERSet compounds that matched phenotypically were on average low.

5. The successful rate of experimental validation is moderate. It does not provide adequate support for the effectiveness of proposed methods.

We find this assertion surprising. Reviewer 3 acknowledges it as a “staggering hit rate” and Reviewer 2 specifically as “a major success of this work.” At 58%, it is about 60x greater than typical HTS hit rates, which are <1% (e.g., see PMID: 32808550). Furthermore, the predictions succeeded despite crossing major species boundaries.

Minor:

1. Page 6, line 150, “Twin-SN” is “Twin-NN”?

Thank you– typo corrected.

2. It will be interesting to see if contrastive learning will perform better than Siamese network.

We regret the ambiguity in our terminology. We have more explicitly clarified how our deep learning approach relates to metric learning and contrastive learning in the text as follows:

Introduction:

“Siamese networks pre-date and often became the neural network models and loss functions subsequently adopted by various studies in deep metric learning and contrastive learning.”

Methods, under “Training deep metric learning models for phenotypic distance”:

“Our Twin-NN and Twin-DN models use a Contrastive Divergence loss function and may be thought of as contrastive learning approaches that simply use real-world (experimentally collected) compound-replicate data instead of relying on the synthetically generated data-augmentation procedures used in contrastive learning architectures such as SimCLR⁶⁶. Thus, our models are formally metric learning models because we provide a training label for each compound pair, reflecting whether the MI trace replicates are of the same small-molecule compound. Conceptually, however, the models are motivated by contrastive learning because they rely on a contrastive logic on replicated observations, albeit without requiring post hoc synthetic augmentations. Thus, a formal contrastive learning model like SimCLR that learns from imperfect synthetic augmentations would be unlikely to perform better than our approach that directly leverages the real-world compound replicate readouts we purposefully collected in a “fit to purpose” way specifically for this study.”

Reviewer #1 (Remarks on code availability):

The codes work as expected.

We are glad to hear it.

Reviewer 2

In this manuscript, Gendelev et al. present a powerful machine learning-based approach to identify and characterize neuroactive drugs. The authors first generated time series of aggregate movement across multiple zebrafish larvae that had been exposed to a sequence of sensory stimuli in the presence or absence of selected small molecules. They then developed a pipeline based on twin neural networks to distinguish specific behavioral changes from normal behavioral profiles. The authors found that the new machine learning-based pipeline performed much better than previously established methods, and the trained models clustered neuroactive compounds in good agreement with their known mechanism of action. They also identified several new compounds that elicited phenotypes similar to previously characterized neuroactive compounds. In vitro data with human targets acquired by an external screening service confirmed many of the new hits and identified compounds with low micromolar binding affinities to their predicted targets, which is a major success of this work. A clever pairwise comparison between the behavioral phenotypes and

corresponding differences in the chemical structures of the drugs is a significant addition to the field. The authors round off the paper with a well-balanced discussion of the limitations of their work. In addition to the software code, the data sets are available from a Zenodo repository as an extra resource for the community. As detailed below, a few minor points should be addressed before publication to improve the clarity of presentation for a broad readership.

We thank the Reviewer for their strong support of the study and for their feedback and suggestions below, which we have adopted.

1) It would be useful to highlight the stimuli that were applied in Fig. 1d, so that readers can interpret the MI changes.

Great point. We have added a detailed stimulus legend to Figure 1d.

2) Please add a y axis definition and scale for figures such as Fig. 1d, Fig. S1 etc.

We thank the Reviewer for the suggestion and have added y-axis definitions and scales into all main-text and supplemental figures containing time series data. In regenerating these MI traces for Figure 7, we also took the opportunity to highlight the compound pair examples that were at further extremes of the quadrants.

3) Please describe in the figure legend what the orange color in Fig. 1e indicates.

We have clarified the coloring in the legend for Figure 1d-e.

4) It would be helpful to improve the clarity of Fig. 3d. As the y axis indicates “Cumulative #Drugs”, it seems that the integral of the curves should be the total number of drugs. The integral is expected to be identical for Twin-NN and Corr, but the areas under the curves differ. Please clarify.

We agree this needed clarification and have revised the legend and text describing Figure 3d. The integral would match the total number of compounds if the computational methods eventually recovered all replicates of all drugs in the study, but they do not because the methods and the data assumptions are not perfect. For instance, some compounds may not affect zebrafish behavior at 10 uM - no method should be able to correctly differentiate its replicates from negative-control wells or those of other behaviorally inactive compounds. All of these cases will decrease the y-axis maximum.

Twin-NN distance nonetheless recovers more replicates across many more drugs than correlation distance (as shown in the Figure). We have added some clarification text in the accompanying legend for Fig 3d as follows:

“We compute an all-by-all distance matrix across NT-650 compounds at the individual replicate level and determine how many replicate wells of the compound appear within the top 50 most similar ranked wells. We plot the cumulative total of unique drugs (y-axis) versus increasing count of identified replicates (x-axis). The y-axis maximum does not reach the total number of NT-650 compounds because neither method perfectly ranks all replicates within the top 50 most phenotypically similar ranked wells for all NT-650 compounds. Indeed, some compounds are inactive, with replicates indistinguishable from DMSO.”

5) In the Results section, please add the data for “not shown” or change the statement.

We have changed the statement as follows: “We considered using recurrent architectures – neural networks designed to operate on sequences, such as the LSTM⁴⁵ or GRU^{45,46} – but were concerned that the long length of the time series samples raised issues with vanishing gradients and run-time.”

6) Some figure references seem to be incorrect in the text (e.g. Fig. 3a-b in the second paragraph of the Results section).

Thank you, good point. We corrected the reference to the correct Figure.

7) It would be useful if the primary data for the results in Fig. 6b could be shown in the Supplementary Material.

We indeed report the underlying primary data for Figure 6b in Supplementary Table T2.

8) The description “We color the dots ...” does not seem to agree with the presentation in Fig. 7. Please clarify.

We agree and have removed the incorrect description.

9) Table T3 contains duplicates (e.g. Clozapine – Chlorpromazine HCl). Please clarify.

Good eye! The NT-650 compound set came plated from the supplier with two distinct Clozapine wells, so Table T3 shows two different examples of Clozapine matched with Chlorpromazine. Chlorpromazine itself is not duplicated and shows up as a unique drug in Table T3. We added some clarifying text to the Results section, as follows: “We note that there are two seemingly identical rows in the table, for Clozapine with Chlorpromazine, since there were two unique occurrences of Clozapine in the NT-650 plates provided by the supplier.”

10) The statement “Individual data points are shown when possible, and always for $n \leq 10$ ” in the Editorial Policy Checklist does not seem to be correct for Fig. S5-13.

We will follow the Editor’s guidance on the journal’s preferred dose-response curve format. Meanwhile, we have indeed replaced Figures S5-13 with variants showing the individual data points as requested.

11) References to other recent papers that have used twin neural networks in zebrafish studies (e.g. PMIDs 35309679, 37996754) could be included.

Thank you for suggesting these references; we have added their citations to the introduction.

Reviewer 3

The manuscript by Gendelev et al. entitled "Deep phenotypic profiling of neuroactive drugs in larval zebrafish" describes a deep learning-assisted approach for characterizing and classifying behavioral changes induced by neuroactive drugs in larval zebrafish. Although the screening approach has been previously reported by the authors, the analysis approach by contrastive learning is novel and significantly outperforms previous correlation distance-based methods. In addition to establishing a drug-induced zebrafish "behaviorome", the authors discovered novel druglike compounds with diverse structure at a staggering hit rate of 58% through an additional screen of uncharacterized compounds followed by validations via in vitro binding assays. The characterization of a shortcut learning phenomenon is also insightful. The study represents a significant advance in phenotypic screening methodology and is suitable for publication in Nature Communications with the following suggested edits.

We thank the Reviewer for their support of the study’s publication as a significant advance.

1: Figure 4: The behaviorome is apparently classified into distinct clusters. Are different clusters enriched in neuroactive drugs belonging to different functional categories? If so, please annotate.

The Reviewer raises an exciting but hard-to-quantify question. We originally attempted to address it qualitatively by visualizing the 14 reference molecules as landmarks in Figure 4, and indeed when drilling down on the clusters, we find intriguing and sensible compound neighbors that in turn often share receptor activity in cross-species prospective testing (e.g., as borne out in Figure 6). But neuroactive compound function is nuanced, with individual molecules delivering different indications in different therapeutic or biological contexts. The frequent polypharmacology or “network pharmacology” seen in neuroactive compounds only exacerbates this clustering challenge. Accordingly, in adopting the Reviewer’s suggestion, we attempted a few approximations of annotating function onto these maps:

1. Anatomical Therapeutic Chemical (ATC) codes

Many of the NT-650 dataset’s neuroactive compounds are not approved drugs. Consequently, we could not map more than a handful of them to formal ATC codes.

2. Vendor plate classes

The NT-650 plates came with 14 supplier-provided plate-wide ligand “class” labels, such as “Opioids” and “Serotonergic Ligands”. Since we had these labels for all 650 compounds, we made a behaviorome plot for the 325 compounds with strong predicted phenotypes according to the random forest drug-vs-DMSO classifier in a new Supplementary Figure S16. We colored the compounds by these general “class labels” and observed weak clustering in the UMAP. For example, we note a cluster of green (Dopaminergic) compounds in the middle left region of the plot (approx UMAP coordinates -4,-1) and a relatively dense population of blue (Adrenergic) compounds in a neighboring region a bit above and to the right (approx UMAP coordinates -2,1).

We did not find this weak clustering surprising. The class labels are broad and do not capture key differences in the mechanisms of action among compounds belonging to the same overall class. For example, many dopaminergic ligands act through specific dopamine receptor subtypes. Furthermore, GPCR ligands can have various levels of activity across many GPCRs. Receptor activity combinations can contribute strongly to compound-induced effects *in vivo*. We believe that the weak clustering reinforces the value of explicit high-throughput behavior profiling in zebrafish combined with metric-learning models to group compounds based on observed *in-vivo* response instead of generic labels. We describe this in the manuscript as follows:

“When we colored the 325-compound subset predicted to have strong phenotypes (Methods) by the generic functional “classes” corresponding to plate-wide assignments by the chemical vendor (e.g., “opioids”, “serotonergic ligands”), we observed only a weak correlation with position in the UMAP (Supplementary Figure S16). This was unsurprising, as these classes do not account for the polypharmacology of many neuroactive compounds.”

2: I would be very interested to see the DIVERset behavioral profiles clustered with the NT-650 dataset in the same UMAP plot as a supplementary figure. I'm wondering if you were able to expand the behaviorome established by the NT650 compounds through behaviors generated by the DIVERset compounds. Were you able to discover behavioral profiles that significantly deviated from the NT650 compounds?

This is an intriguing idea. While we had used NT-650 compounds as “queries” to computationally search the DIVERSet screen dataset for novel compound behavioral matches (e.g., Figure 6), we had not previously constructed a joint UMAP visualization. We have now done so, as the new Supplementary Figure S17. As in the original behaviorome (Figure 4), we have marked the 14 selected experimentally tested NT650 compounds and DMSO. We see a substantial learned-distance overlap between the behavioral profiles encountered in the two screens despite their chemical structural diversity (e.g., Figures S14-S15). However, some UMAP regions have a higher density of NT-650 compounds and vice-versa, suggesting certain behaviors may more commonly appear in one screen than the other.

We have added text to the “Model generalization and novel drug discovery” subsection of Results:

“We also asked how the phenotypic space of the DIVERSet compares to the NT-650. We computed a combined UMAP (all compounds from NT-650 combined with DIVERSet), colored by the dataset the compounds came from (Supplementary Figure S17). Strikingly, there is a large overlap between the blue (NT-650) and orange (DIVERSet) compounds, considering that the models were not trained on DIVERSet data. Despite this out-of-domain setting, on new compounds and behavioral data, the models usefully associate novel compounds from the DIVERSet with known compounds in the NT-650 set. This overlap supports the phenosearch approach, as most NT-650 query compounds have many phenotypically-similar DIVERSet novel compound neighbors in the map. Some regions have a higher density of NT-650 compounds and vice-versa, suggesting behaviors more commonly appear in one screen versus the other. Further studies might focus on selected sub-regions with DIVERSet density higher than NT650 to explore potentially novel phenotypic space.”

3: The same drug can elicit different behavior at different doses, and the correlation between dose and behavior changes might not be linear. I understand the decision to focus on one dose for the initial platform

setup, but would be very interested to see how this model generalizes to different doses of the same drug, such as by testing different doses of a few selected drugs and see how they rank by Twin-NN distance among other drugs screened. I would also be interested to see a discussion of whether future updates of the model can be trained with different drug doses.

This is also an interesting suggestion, but we could not test it in a timely way due to personnel and funding limitations. Indeed, as compound identity and chemical structure are not model inputs, there is no reason a priori that we could not inject any MI trace (corresponding to dose, co-dosed compounds, environmental perturbations, non-WT fish, etc) and use the model's inference capabilities to place the MI within the landscape (or outside of it!). We have noted this in the Discussion as follows:

“Another caveat is that due to experimental scope, we could not account for compound dose, as all compounds were screened at a single concentration of 10uM. However, models trained on such data could presumably infer phenotypes triggered by alternative doses of compounds, as they already distinguish DMSO wells from treatments. In the out-of-domain application of the models to the larger DIVERSet library, compounds populated the full gamut of behavioral space compared to compounds in the NT-650 training set, including those that were fully or partially inactive. In future directions, the models might serve as bridges to relate otherwise disparate perturbations, ranging from compound doses to functional genomics or the role of environmental changes.”

4: The paper PMID:36814839 describing another deep learning-assisted behavioral classification method in zebrafish is relevant and should be included in the reference.

Thank you; we have added the citation to the introduction.

Author-Initiated Edits

In responding to Reviewer 2's request for clarified axis labels, we noted that the previous text described the MI traces as recording frames every 1/30th of a second. This referred to an earlier version of assay equipment using a camera with a slower framerate. We have corrected it to the faster 1/100th of a second and updated the corresponding descriptions, including longer trace lengths (101,250 frames/trace) across the manuscript.

Response to Reviewer Comments

Reviewer 3

The authors have thoroughly addressed all my previous comments. I have no further comments and recommend the manuscript to be accepted for publication.

Thank you - we have appreciated the feedback and are delighted to hear this.